EMBO
Molecular Medicine

# Sprouty2 loss-induced IL6 drives castration-resistant prostate cancer through scavenger receptor B1

Rachana Patel[1,*] (iD), Janis Fleming[1], Ernest Mui[2], Carolyn Loveridge[2], Peter Repiscak[2], Arnaud Blomme[1], Victoria Harle[2], Mark Salji[2], Imran Ahmad[1,2], Katy Teo[2], Freddie C Hamdy[3], Ann Hedley[1], Niels van den Broek[1], Gillian Mackay[1], Joanne Edwards[2], Owen J Sansom[1] (iD) & Hing Y Leung[1,2,**] (iD)

## Abstract

Metastatic castration-resistant prostate cancer (mCRPC) is a lethal form of treatment-resistant prostate cancer and poses significant therapeutic challenges. Deregulated receptor tyrosine kinase (RTK) signalling mediated by loss of tumour suppressor Sprouty2 (SPRY2) is associated with treatment resistance. Using pre-clinical human and murine mCRPC models, we show that SPRY2 deficiency leads to an androgen self-sufficient form of CRPC. Mechanistically, HER2-IL6 signalling axis enhances the expression of androgen biosynthetic enzyme HSD3B1 and increases SRB1-mediated cholesterol uptake in SPRY2-deficient tumours. Systemically, IL6 elevated the levels of circulating cholesterol by inducing host adipose lipolysis and hepatic cholesterol biosynthesis. SPRY2-deficient CRPC is dependent on cholesterol bioavailability and SRB1-mediated tumoral cholesterol uptake for androgen biosynthesis. Importantly, treatment with ITX5061, a clinically safe SRB1 antagonist, decreased treatment resistance. Our results indicate that cholesterol transport blockade may be effective against SPRY2-deficient CRPC.

**Keywords** androgen receptor; cholesterol; interleukin 6; prostate cancer; scavenger receptor B1

**Subject Categories** Cancer; Pharmacology & Drug Discovery

## Introduction

Prostate cancer is the most common malignancy in men. Treatment resistance and cancer dissemination are the major causes of disease-associated mortalities in most advanced cancers, including prostate cancer. Androgens and androgen receptor (AR) are essential for growth and survival of prostate cancer (Yap et al, 2016). Androgen deprivation therapy (ADT) administered through surgical or medical castration remains the current initial treatment for metastatic prostate cancer. Despite an initial favourable response to ADT (even when combined with upfront docetaxel chemotherapy), a majority of the patients will progress to an advanced disease state termed castration-resistant prostate cancer (CRPC). Although a subset of the emerging CRPC may exhibit AR-independent characteristics, AR reactivation remains as a significant feature of advanced disease (Watson et al, 2015).

Experimental and clinical investigations have identified several mechanisms that maintain AR activity through amplifications, mutations and splicing of AR along with tumoral androgen biosynthesis (Watson et al, 2015). The clinical success of abiraterone, an androgen biosynthesis inhibitor, in improving the treatment response highlights the importance of androgens in the emergence of CRPC (James et al, 2017). Recent translational studies have also revealed the causative role of HSD3B1, an androgen biosynthetic enzyme, in the development of CRPC by intra-tumoral androgen biosynthesis from cholesterol (Chang et al, 2013; Hearn et al, 2016). Studies implicating the association of tumoral cholesterol biosynthesis and cholesterol uptake by lipoprotein receptors such as low-density lipoprotein receptor (LDLR) and scavenger receptor B1 (SRB1) in the development of CRPC have proposed a potential role for cholesterol in prostate cancer progression (Kim et al, 2014; Schorghofer et al, 2015; Furuya et al, 2016). Besides local tumour factors, systemic physiological conditions such as dyslipidaemia and chronic inflammation may also govern treatment response and disease progression (Fearon et al, 2012). These factors are particularly important in prostate cancer as both systemic cholesterol homeostasis and pro-inflammatory cytokines such as IL6 have been implicated in CRPC (Rice et al, 2012; Culig, 2014; Divella et al, 2016). Therefore, studying the treatment response at a systemic level may improve our current understanding of disease progression.

1 Cancer Research UK Beatson Institute, Glasgow, UK
2 Institute of Cancer Sciences, Glasgow, UK
3 Nuffield Department of Surgical Sciences, John Radcliffe Hospital, University of Oxford, Headington, Oxford, UK
*Corresponding author. Tel: +44 141 330 8170; E-mail: r.patel@beatson.gla.ac.uk
**Corresponding author. Tel: +44 141 330 3658; E-mail: h.leung@beatson.gla.ac.uk

Comprehensive genomic and transcriptomic profiling revealed activation of receptor tyrosine kinase (RTK)-mediated oncogenic (RAS/ERK and PI3K/AKT) signalling pathways, coupled with inactivation of tumour suppressors such as the lipid phosphatase PTEN and the RAS/ERK pathway inhibitor Sprouty2 (SPRY2), in advanced prostate cancer (Taylor et al, 2010). These oncogenic signalling cascades can mediate treatment resistance in both AR-dependent and independent manner (Mulholland et al, 2011; Rick et al, 2012). Experimental models of prostate cancer have identified PTEN and SPRY2 as important molecular determinants of aggressive prostate carcinogenesis including metastases (Ahmad et al, 2011, 2013; Schutzman & Martin, 2012; Patel et al, 2013; Assinder et al, 2015). While PTEN loss has been implicated in driving both AR-dependent and independent forms of CRPCs (Mulholland et al, 2011; Yang et al, 2018), the functional contributions of SPRY2 deficiency in promoting treatment resistance remain undefined.

Using murine pre-clinical models of prostate cancer, we show how SPRY2 deficiency leads to treatment resistance. SPRY2 deficiency causes CRPC by inducing enhanced cholesterol uptake through the HER2-IL6 signalling axis and tumoral androgen biosynthesis. By studying CRPC at an organism level, we have identified systemic inflammation and cholesterol homeostasis as important processes influencing the treatment response. Hence, building on the extensive genomic profiles of clinical prostate cancer in the literature, we report on mechanistic insight into molecular determinants that govern disease progression to CRPC.

## Results

### SPRY2 deficiency mediates CRPC

To evaluate the clinical significance of SPRY2 deficiency in governing treatment response, we investigated the prognosis of ADT-treated prostate cancer patients based on the tumour levels of SPRY2. Prostate cancer patients with low SPRY2 levels, defined as below median histoscore, showed significantly lower relapse-free and overall survival following ADT (Fig 1A and B). Similarly, patients with low SPRY2 expression in TCGA prostate cancer dataset also have significantly reduced relapse-free survival (Fig EV1A; Cancer Genome Atlas Research, 2015). We next investigated SPRY2 status in hormone-naïve (HN) and CRPC matched tissue microarray comprising of prostate tumour biopsy pairs at diagnosis and after recurrence from the same individuals. SPRY2 levels were significantly lower in CRPC compared to matched HNPC (Fig 1C and D). PTEN immunoreactivity in HN and CRPC paired tumour samples was similar, with almost 50% of the HN and CRPC matched cases showing no detectable PTEN expression (Fig EV1B). Consistent with our previous reports on co-occurrence of SPRY2 and PTEN alterations in prostate cancer (Gao et al, 2012; Patel et al, 2013), SPRY2 and PTEN levels significantly correlated in CRPC (Fig EV1C). Since SPRY2 expression was progressively lost from HN state to CRPC, we hypothesised that SPRY2 deficiency may modulate tumour response to ADT.

To investigate if SPRY2 and PTEN deficiencies may mediate CRPC, we used the genetically engineered mouse model of prostate cancer, Nkx3.1 Pten$^{fl/+}$ Spry2$^{fl/+}$ (hereafter referred as NPS) (Gao et al, 2012). In the NPS model, the Nkx3.1 promoter-driven Cre recombinase mediates heterozygous loss of Pten and Spry2 in the AR-proficient luminal cells of the prostatic epithelium. The concomitant heterozygous loss of Pten and Spry2 led to significant prostate tumour burden after 50 weeks with nuclear AR, p-AKT and p-ERK1/2 (Figs 1E and EV1D). Despite an initial favourable response to ADT (achieved by castration), dual heterozygous loss of Pten and Spry2 led to CRPC, signified by comparable tumour burden and overall survival in mock and ADT-treated mice (Figs 1F and G, and EV1D–F). CRPC arising from the NPS model exhibited sustained nuclear AR, p-AKT and p–ERK1/2 (Figs 1E and EV1G). We have previously demonstrated that prostate-specific heterozygous loss of Spry2 does not result in tumour formation, while the heterozygous loss of Pten alone causes PIN (prostatic intraepithelial neoplasia) lesions (Gao et al, 2012; Patel et al, 2013). In this study, the response to ADT was found to be similar across wild-type, Spry2 heterozygous and Pten heterozygous mice (Fig EV1H–J). We next investigated the effects of SPRY2 on growth under hormone deprivation in LNCaP cells, an AR-proficient prostate cancer cell line deficient for both SPRY2 and PTEN (Fig EV1K). Consistent with our previous observations, SPRY2 overexpression significantly decreased the growth of LNCaP cells in hormone-proficient conditions (FBS) (Figs 1H and EV2A). Importantly, the SPRY2 overexpression-induced inhibition of cell growth was significantly more profound under hormone-deprived culture condition (Fig 1H).

To explore if SPRY2 could mediate resistance to ADT independent of PTEN status, we investigated the levels of AR, PTEN and SPRY2 in a panel of prostate cancer cell lines (Fig EV1K). AR- and PTEN-proficient prostate cancer cell lines (CWR22Res, CWR22RV1 and VCaP) showed significantly enhanced expression of AR variants including ARV7 (Figs EV1K and EV2B). Unlike CWR22RV1 cells, a CRPC variant cell line (Sramkoski et al, 1999; Tepper et al, 2002), the CWR22Res cells (Nagabhushan et al, 1996) retain their hormone responsiveness and show a significant decrease in growth rate under hormone-deprived conditions (Fig EV2C). It is worth noting that CWR22RV1 expressed lower levels of SPRY2 than CWR22Res cells. To further investigate the potential role of SPRY2 in mediating CRPC, we stably knocked down SPRY2 in CWR22Res cells (Fig EV2D). Compared to the non-silencing vector control cells (Nsi), the stable knockdown (KD) of SPRY2 (CL3 and Pool) in CWR22Res cells did not alter the growth rate in hormone-proficient conditions (Fig 1I). Importantly, SPRY2-depleted CWR22Res cells showed sustained growth rate even in hormone-deprived condition (Fig 1I). In addition, we developed a murine prostate orthograft model using CWR22Res cells. We surgically implanted CWR22Res cells in one of the anterior prostate lobes in CD-1 nude mice. When the prostate orthografts were either palpable or detectable by ultrasound imaging (typically at 30 days following implantation), the mice were randomised to receive either ADT (by castration) or mock treatment (sham surgery) (Fig EV2E and F). As expected, following ADT, CWR22Res orthografts showed a significant reduction in growth rate compared to mock treatment (Fig EV2E and F). We next studied CWR22Res SPRY2 KD generated orthografts to investigate the role of SPRY2 in tumour response to ADT (Fig EV2G). All mock-treated animals achieved maximum permitted tumour burden around 73 days post-implantation irrespective of SPRY2 status (Fig EV2H). While ADT-treated mice with Nsi orthografts survived longer than mock-treated mice, ADT-treated mice with SPRY2-deficient orthografts showed signs of weight loss

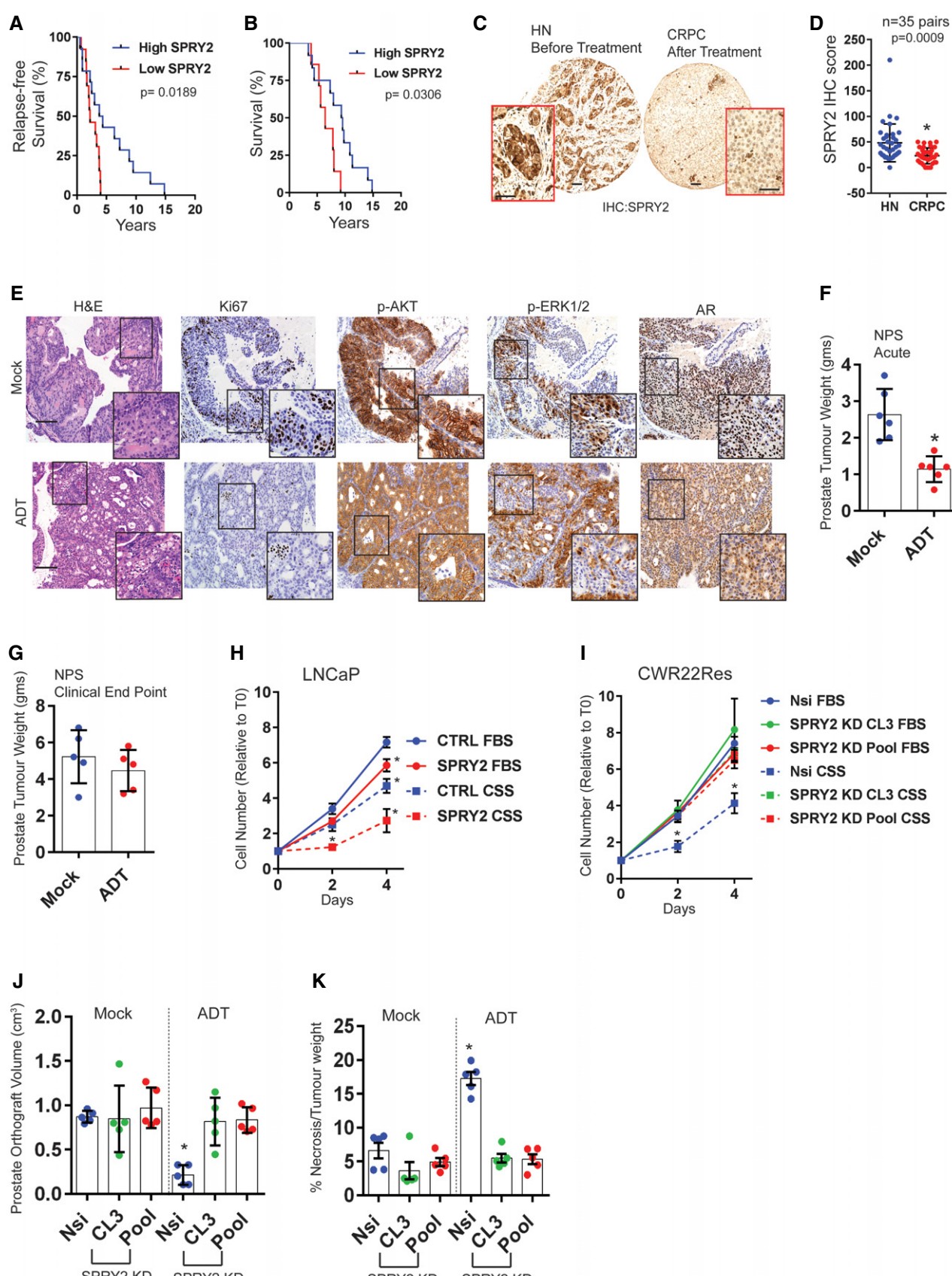

**Figure 1.**

**Figure 1.  SPRY2 deficiency leads to CRPC.**

A   Kaplan–Meier plot for the relapse-free survival of prostate cancer patients treated with androgen deprivation therapy (ADT). SPRY2-high patients, $n$ = 14; SPRY2-low patients, $n$ = 13; log-rank (Mantel–Cox) test; $P$ = 0.0189.

B   Kaplan–Meier plot for overall (post-diagnosis) survival of ADT-treated prostate cancer patients. SPRY2-high patients, $n$ = 12; SPRY2-low patients, $n$ = 7; log-rank (Mantel–Cox) test; $P$ = 0.0306.

C   Representative images for immunostaining for SPRY2 in HN (hormone-naïve) and CRPC (castration-resistant prostate cancer) matched clinical prostate cancer sections. Scale bars = 100 μm.

D   Immunostaining for SPRY2 in HN and CRPC matched clinical prostate cancer sections represented as IHC (immunohistochemistry) scores ($n$ = 35 pairs; paired two-tailed Student's $t$-test; *$P$ = 0.0009).

E   Representative H&E and immunostained images of prostate sections from *Nkx3.1 Pten*$^{fl/+}$ *Spry2*$^{fl/+}$ (NPS) mice. Scale bars = 100 μm.

F   Prostate tumour burden 1 month post-ADT in NPS mice ($n$ = 6 mice per group; unpaired two-tailed Student's $t$-test; *$P$ = 0.0009).

G   Prostate tumour burden at clinical endpoint in NPS mice ($n$ = 5 mice per group).

H, I   Growth rate of LNCaP control and SPRY2-overexpressing cells (H); and CWR22Res Nsi (non-silencing) control and CL3 and Pool stable SPRY2 knockdown cell lines (I) relative to T0 (Day 0) in hormone-proficient (FBS) medium and hormone-depleted (CSS) medium ($n$ = 3; *$P$ < 0.05; ANOVA Tukey's test).

J   CWR22Res prostate orthograft tumour volumes from 60-days timed experiment ($n$ = 5 mice per group; *$P$ < 0.05; ANOVA Tukey's test).

K   The effects of ADT on the CWR-22 prostate orthografts from 60-days timed experiment represented as % necrosis/tumour weight ($n$ = 5 mice per group; *$P$ < 0.05; ANOVA Tukey's test).

Data Information: In (D, F, G, J, K), each data point represents one independent observation. In (H, I), each data point is mean of three independent observations. In (D, F, G–K), data are presented as mean ± SD.

around 60 days (Fig EV2H and I). The ADT-treated mice with SPRY2-deficient orthografts reached clinical endpoints between 60 and 70 days (Fig EV2H). Hence, we used a refined 60-days timed experimental protocol for subsequent investigations. The ADT-treated Nsi orthografts showed a significant reduction in orthograft volume with increased ADT-induced tumour necrosis when compared to mock-treated Nsi orthografts (Figs 1J and K, and EV2J and K). Compared to Nsi orthografts, SPRY2-deficient orthografts showed sustained growth under ADT conditions with minimal ADT-induced tumour necrosis (Figs 1J and K, and EV2J). Since CWR22Res cells express AR-V7 which may mediate CRPC progression, we next investigated the effects of SPRY2 KD on AR expression (Watson *et al*, 2015). SPRY2 deficiency did not alter the expression of AR (full length and V7; Fig EV2L).

The PTEN levels remained unaltered in all CWR22Res orthografts (Fig EV2M). Mimicking the NPS model and LNCaP cells, by transiently knocking down PTEN, we show that SPRY2 deficiency provided a survival advantage under hormone-deprived conditions independent of PTEN expression (Fig EV2N and O). Thus, SPRY2 deficiency may mediate resistance to ADT.

**SPRY2 deficiency leads to androgen self-sufficient form of CRPC**

To characterise SPRY2 deficiency-mediated ADT resistance, we examined the status of AR in our orthografts. In response to ADT, hormone-responsive Nsi orthografts showed decreased levels of AR (full length and variants) and prostate-specific antigen (PSA), an AR target gene expression (Figs 2A and B, and EV3A and B). In contrast, SPRY2-deficient prostate orthografts maintained their AR nuclear localisation, total levels of AR (full length and variants) and PSA expression despite ADT (Figs 2A and B, and EV3A and B). We further performed gene set enrichment analyses (GSEA) on microarray-based transcriptomic profiles of CWR22Res Nsi control and SPRY2 KD Pool cells. In GSEA, SPRY2-deficient cells showed significant enrichment of steroid biosynthetic genes (Fig 2C). Together, these observations suggested a potential role of SPRY2 deficiency in mediating a type of CRPC with sustained AR activity.

Tumoral androgen biosynthesis may mediate the development of CRPC with active AR pathway (Watson *et al*, 2015). To explore this,

we examined the androgen biosynthetic pathway in SPRY2-deficient CRPC (Fig 2D). SPRY2-deficient orthografts in ADT-treated mice maintained their intra-tumoral testosterone levels, despite castrated levels of circulating testosterone (Fig 2E and F). Furthermore, only under ADT conditions, SPRY2-deficient orthografts showed elevated levels of tumoral cholesterol, an important precursor for androgen biosynthesis (Fig 2G). Amongst the key androgen biosynthetic enzymes [C17,20-lyase (CYP17A1), 3 beta-hydroxysteroid dehydrogenase (HSD3B1) and 17 beta-hydroxysteroid dehydrogenase (HSD17B1)], HSD3B1 expression was significantly increased in SPRY2 KD CWR22Res cells (Fig EV3C and D). While CYP17A1 expression remained unaltered in orthografts, SPRY2-deficient prostate orthografts showed a significant increase in HSD3B1 expression in response to ADT (Figs 2A and H, and EV3E). The ADT-resistant NPS tumours also showed elevated levels of HSD3B1 (Fig EV3F). Furthermore, ectopic SPRY2 expression in LNCaP cells significantly decreased expression of both CYP17A1 and HSD3B1 (Figs 2I and EV3G). Similarly, compared to untreated VCaP (PTEN proficient and SPRY2 deficient) prostate orthografts, tumoral HSD3B1 expression was significantly increased following ADT (Fig EV3H).

We next tested the sensitivity of SPRY2-deficient cells to the CYP17A1 inhibitor abiraterone, which improves the treatment response of prostate cancer patients when combined with standard-of-care ADT (James *et al*, 2017). Abiraterone showed comparable growth inhibition in control and SPRY2-expressing LNCaP cells in both hormone-proficient and hormone-deficient conditions (Fig EV3I). Interestingly, in hormone-deficient conditions, CWR22Res SPRY2 KD cells were less sensitive to abiraterone treatment compared to Nsi controls (Fig EV3J). Since increased HSD3B1 expression can mediate abiraterone resistance (Chang *et al*, 2013), we next explored the importance of HSD3B1 in ADT-resistant SPRY2 deficient cells. In CWR22Res cells, HSD3B1 knockout decreased the growth rate of SPRY2-deficient cells in hormone-deprived conditions with an associated decrease in the cellular testosterone levels (Figs 2J and EV3K–M). Furthermore, HSD3B1 knockout enhanced the sensitivity of ADT-resistant SPRY2-deficient cells to abiraterone treatment (Fig 2J).

We have previously demonstrated that SPRY2 deficiency promotes prostate carcinogenesis through aberrant HER2 signalling

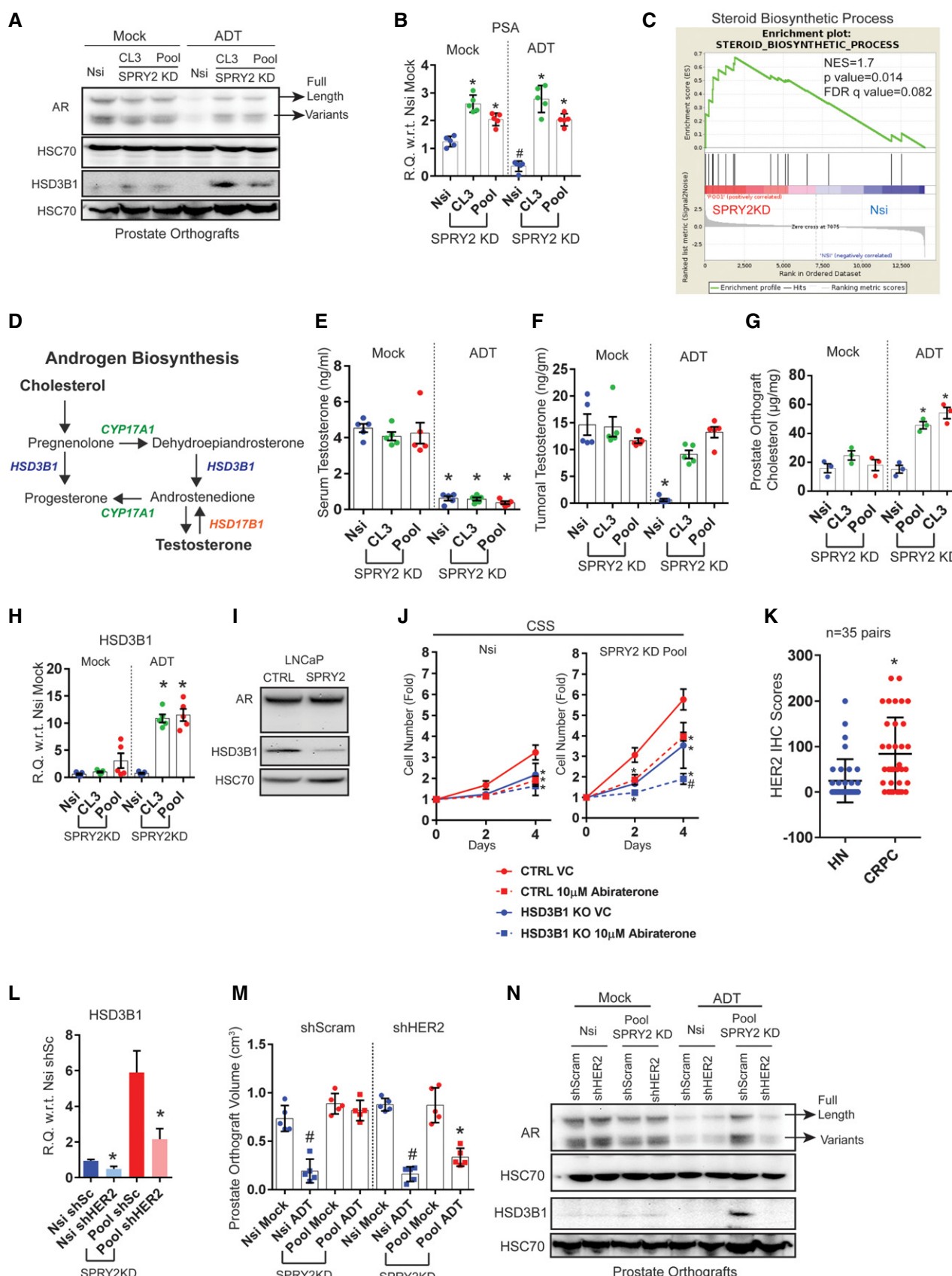

**Figure 2.**

**Figure 2. SPRY2 deficiency mediates androgen autonomous CRPC.**

A    Representative immunoblot images for indicated proteins in lysates from indicated CWR22Res orthografts from 60-days timed experiment. HSC70 is used a loading control.

B    Relative quantitation of PSA mRNA in CWR22Res prostate orthografts from 60-days timed experiment ($n$ = 5 mice per group; [#]$P < 0.05$ compared to Nsi mock; *$P < 0.05$ compared to respective Nsi treatment; ANOVA Tukey's test).

C    GSEA analysis showing steroid biosynthetic process gene enrichment in CWR22Res SPRY2-deficient (Pool) cells compared to Nsi control cells.

D    Schematic presentation of androgen biosynthesis pathway.

E, F    Serum (E) and intra-tumoral (F) testosterone from mice bearing indicated orthografts from 60-days timed experiment measured using ELISA ($n$ = 5 mice per group; *$P < 0.05$; ANOVA Tukey's test).

G    Total cholesterol levels in indicated prostate orthografts from 60-days timed experiment ($n$ = 3 mice per group; *$P < 0.05$; ANOVA Tukey's test).

H    Relative quantitation of human 3-β-hydroxysteroid dehydrogenase (HSD3B1) mRNA in CWR22Res prostate orthografts from 60-days timed experiment ($n$ = 5 mice per group; *$P < 0.05$; ANOVA Tukey's test).

I    Representative immunoblot images for indicated proteins in lysates from control and SPRY2-expressing LNCaP cells. HSC70 is used a loading control.

J    Growth rate of HSD3B1 KO CWR22Res Nsi control and Pool stable SPRY2 KD cells relative to T0 in hormone-depleted (CSS) medium with indicated treatments ($n$ = 3; *$P < 0.05$ compared to CTRL VC at the indicated time; [#]$P < 0.05$ compared to all treatments at the indicated time; ANOVA Tukey's test).

K    Immunostaining for HER2 in HN and CRPC matched clinical prostate cancer sections represented as IHC scores ($n$ = 35 pairs; *$P < 0.0001$; paired two-tailed Student's $t$-test).

L    Relative quantitation of HSD3B1 mRNA in Nsi and Pool (SPRY2 KD) CWR22Res prostate cancer cells with stable expression of shSc (shScram) or shHER2 ($n$ = 3; *$P < 0.05$—shHER2 compared to respective shSc; unpaired two-tailed Student's $t$-test).

M    The effects of HER2 knockdown on CWR22Res orthografts represented as tumour volume ($n$ = 5 mice per group; [#]$P < 0.05$ for Nsi orthografts; *$P < 0.05$ for Pool orthografts; ANOVA Tukey's test).

N    Representative immunoblot images for indicated proteins in lysates from indicated CWR22Res orthografts from 60-days timed experiment. HSC70 is used a loading control.

Data Information: In (B, E–H, K, M), each data point represents one independent observation. In (J), each data point is mean of three independent observations. In (B, E–H, J–M), the data are presented as mean ± SD.

Source data are available online for this figure.

(Gao *et al*, 2012). Compared to matched hormone-naïve clinical PC, HER2 immunoreactivity was significantly higher in CRPC (Fig 2K). CWR22RV1 cells (CRPC variant of CWR22 cells) also showed decreased levels of SPRY2 with increased levels of HSD3B1, HER2 and EGFR (Figs EV1K and EV3N). To test the role of HER2 in mediating SPRY2 deficiency-induced treatment resistance, we stably knocked down HER2 expression in Nsi and SPRY2 KD (Pool) CWR22Res cells (Fig EV3O). HER2 KD significantly decreased HSD3B1 expression and sensitised SPRY2-deficient orthografts to ADT (Fig 2L–N), along with reduced AR expression (Fig 2N). Thus, SPRY2 deficiency may facilitate treatment resistance by conferring androgen self-sufficiency through HER-mediated induction of HSD3B1.

## SPRY2 deficiency-induced IL6 drives CRPC by elevating tumoral HSD3B1 and cholesterol levels

We next sought to understand the underlying mechanisms of HER2-driven treatment resistance in SPRY2-deficient CRPC. Since stress-induced epithelial cytokines can facilitate androgen biosynthesis (Chun *et al*, 2009), we profiled a panel of cytokines in the orthografts. Using a cytokine array, we identified IL6 as the most elevated cytokine in the SPRY2-deficient orthografts (Fig EV4A). Consistent with the data from cytokine profile, the SPRY2-deficient ADT-resistant orthografts also showed elevated IL6 expression (Fig 3A). HER2 KD diminished the IL6 expression in SPRY2-deficient ADT-resistant orthografts (Fig 3B). Similarly, ectopic expression of SPRY2 in LNCaP cells decreased the IL6 expression (Fig EV4B).

We have previously shown that p38 MAP kinase is an important mediator of HER2 activation following SPRY2 loss (Gao *et al*, 2012). Consistent with this, we find elevated p-p38 levels in SPRY2-deficient CWR22Res cells and p38 inhibitor treatment decreased IL6 and HSD3B1 levels (Figs 3C and EV4C). In clinical prostate cancer samples, HER2 levels significantly correlated with IL6 levels

($r$ = 0.2446; $P$ = 0.0288; $n$ = 80). In clinical CRPC, IL6 staining was significantly increased compared to the matched hormone-naïve (HN) samples (Figs 3D and EV4D). Levels of SPRY2 and IL6 showed a significant inverse correlation in ADT-treated prostate cancer patients with evidence of biochemical relapse ($r$ = −0.427; $P$ = 0.0261; $n$ = 27; Fig EV4E). These observations may suggest an enrichment of tumoral IL6 in a subset of CRPC samples with SPRY2 deficiency. Overall, androgen-independent CWR22RV1 and DU145 prostate cancer cells showed higher levels of IL6 expression compared to hormone-responsive LNCaP and CWR22Res cells (Fig EV4F).

Exogenous IL6 treatment increased HSD3B1 expression in CWR22Res cells, and antibody-mediated IL6 neutralisation significantly decreased both IL6 and HSD3B1 expression in SPRY2 deficient cells (Fig EV4G and H). Thus, SPRY2 deficiency may mediate CRPC by inducing HSD3B1 through HER2-IL6 cytokine axis. To test the therapeutic impact of targeting IL6 signalling, we treated CWR22Res orthograft-bearing mice with tocilizumab, an IL6 receptor antagonist. Tocilizumab treatment significantly sensitised the SPRY2-deficient castration-resistant orthografts to ADT (Fig 3E). Tocilizumab treatment significantly decreased the expression of HSD3B1, PSA and AR levels in SPRY2-deficient orthografts following ADT (Figs 3F and G, and EV4I–K). Tocilizumab treatment also suppressed IL6 expression, suggesting an IL6-driven feed-forward autocrine loop within the orthografts (Fig 3H). We next studied the effects of ADT and anti-IL6 combination therapy in NPS mice. The addition of anti-IL6 treatment significantly sensitised NPS prostate tumours to ADT, with significantly decreased tumoral AR (Figs 3I and EV4L and M). In both orthograft and NPS models, IL6-neutralising therapies did not alter the tumour burden in mock-treated animals, indicating the importance of IL6 in tumour survival in the context of ADT. Since in prostate cancer patients, treatment resistance is frequently associated with enhanced risk of cancer metastases, we characterised the incidence of visceral metastases in

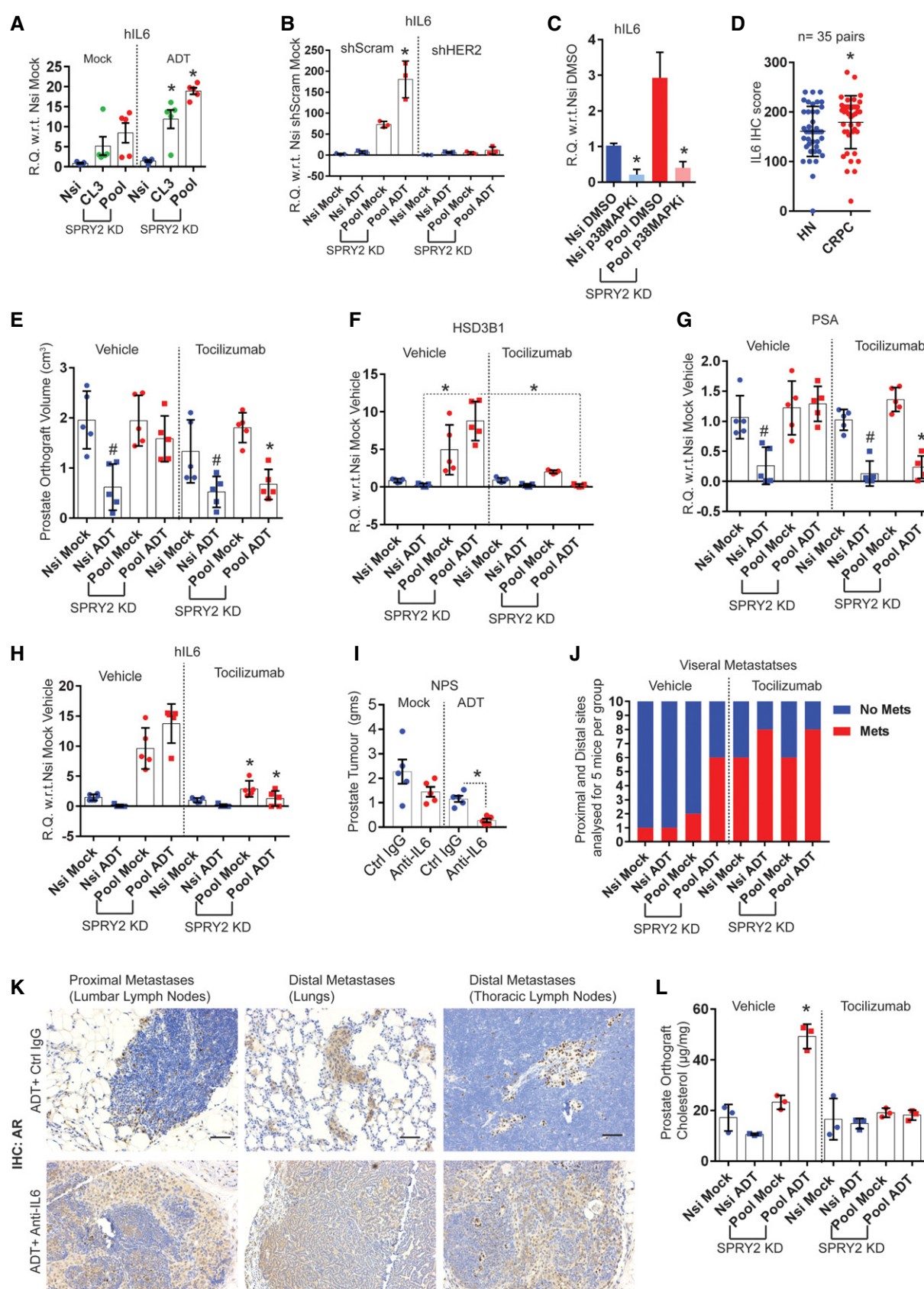

**Figure 3.**

**Figure 3.  SPRY2 deficiency-induced IL6 drives CRPC.**

A, B   Relative quantitation of human IL6 mRNA in the indicated CWR22Res prostate orthografts ($n$ = 5 for A and $n$ = 3 for B mice per group; *$P$ < 0.05 ANOVA Tukey's test).

C      Relative quantitation of IL6 mRNA in Nsi and SPRY2 KD (Pool) CWR22Res cells treated for 16 h with the p38 inhibitor SB203580 (20 μM) ($n$ = 3; *$P$ < 0.05 compared to respective DMSO control; unpaired two-tailed Student's $t$-test).

D      Immunostaining for IL6 in HN and CRPC matched clinical prostate cancer sections represented as IHC scores ($n$ = 35 pairs; *$P$ = 0.05 paired two-tailed Student's $t$-test).

E      The effects of tocilizumab treatment on CWR22Res orthografts represented as tumour volume ($n$ = 5 mice per group; #$P$ < 0.05 for Nsi orthografts; *$P$ < 0.05 for Pool orthografts; ANOVA Dunnett's test).

F–H    Relative quantitation of human (F) HSD3B1, (G) PSA and (H) IL6 mRNA in CWR22Res prostate orthografts ($n$ = 5 mice per group; #$P$ < 0.05 for Nsi orthografts; *$P$ < 0.05 for Pool orthografts; ANOVA Tukey's test).

I      Prostate tumour burden in mock- or ADT-treated NPS mice treated for 1 month with anti-IL6 or anti-IgG (control antibody) ($n$ = 5 mice per group; *$P$ < 0.05 unpaired two-tailed Student's $t$-test).

J      Cumulative visceral (proximal and distal as shown in Appendix Fig S1A) metastases incidence analysed from 10 sites per group (two sites: proximal and distal × 5 mice per group) in the mice with indicated CWR22Res orthografts. Chi-square test: $P$ < 0.001.

K      Representative AR immunostained images as indicated from NPS mice ($n$ = 3). Scale bars = 10 μm.

L      Total cholesterol levels in the indicated CWR22Res prostate orthografts ($n$ = 3 mice per group; *$P$ < 0.05 ANOVA Tukey's test).

Data Information: In (A, B, D, E–I, L), each data point represents one independent observation. In (J), data presented as contingency graph. In (A–I, L), the data are presented as mean ± SD.

CWR22Res prostate orthograft and NPS models. We quantified the incidence of proximal and distal visceral metastasis in each animal based on histological examination and AR staining (Appendix Fig S1A and B). The cumulative analyses of visceral metastases showed increased metastatic incidence in mice bearing SPRY2-deficient orthografts following ADT (Fig 3J). Surprisingly, tocilizumab treatment increased the incidence of visceral metastases irrespective of SPRY2 status, despite enhanced tumour response to ADT (Fig 3J). Similarly, NPS mice treated with combined ADT and IL6-neutralising antibody showed an increase in proximal and distant metastatic incidence and burden (Fig 3K, Appendix Fig S1C–E). These observations indicated that despite the importance of IL6 in driving CRPC, systemic IL6 might play a role in governing metastatic spread. These observations are consistent with a previously reported role of IL6 in restraining the metastatic spread of Pten-deficient prostate cancer (Pencik et al, 2015).

In addition to the effects on AR and HSD3B1, tocilizumab treatment also normalised tumoral cholesterol levels in SPRY2-deficient orthografts following ADT (Fig 3L). Hence, we hypothesised that elevated intra-tumoral cholesterol levels may play a substantial role in the emergence of treatment resistance and strategies to decrease tumoral cholesterol may impact on CRPC.

## SPRY2-deficient tumour-induced IL6 increases systemic cholesterol levels

The increased intra-tumoral cholesterol in SPRY2-deficient orthografts following ADT is unlikely a result of enhanced tumoral cholesterol synthesis since the expression of HMGCR (a rate-limiting enzyme for cholesterol biosynthesis) was significantly lower in the ADT-treated prostate orthografts (Fig 4A). Interestingly, the serum cholesterol levels were significantly higher in castrated mice bearing SPRY2-deficient orthografts (Fig 4B). The elevated serum cholesterol levels may serve as a potential source of cholesterol in ADT-resistant orthografts. Overall, all ADT-treated mice bearing prostate orthografts, irrespective of SPRY2 status, showed elevated serum triglyceride levels (Appendix Fig S2A).

Along with altered circulating cholesterol levels, in mice bearing SPRY2-deficient tumours, serum IL6 levels were also significantly raised following ADT (Fig 4C). Importantly, tocilizumab treatment normalised the circulating levels of cholesterol and IL6 (Fig 4D and E). Besides tumour-derived IL6, IL6 produced by the host adipose tissue may also, at least in part, contribute to the observed systemic increase in IL6 level (Appendix Fig S2B and C). Ectopic IL6 stimulation of cultured adipocytes increased their IL6 expression, suggesting the possibility of an autocrine feed-forward loop in the production of IL6 by adipocytes (Appendix Fig S2D). Our observations indicated a potential role of IL6 in elevating circulating cholesterol, which may serve as a vital source of tumoral cholesterol. We next investigated the association between CRPC and IL6 status in a clinical cohort. CRPC patients have elevated serum IL6 levels when compared to patients with untreated hormone-naïve (HN) prostate cancer (Fig 4F). Serum IL6 levels also significantly correlated with serum PSA levels (Fig 4G). Importantly, CRPC patients with high serum IL6 levels had poorer survival outcome when compared to CRPC patients with low levels of serum IL6 (Fig 4H).

Since tumour-induced IL6 can induce dyslipidaemia (Flint et al, 2016), we next investigated the role of systemic IL6 on host lipid metabolism in our models. Indicative of cancer cachexia, mice harbouring SPRY2-deficient CRPC showed significant loss of epididymal adipose tissue (Fig 4I, Appendix Fig S2E). Since tocilizumab treatment rescued this effect (Fig 4J), we further investigated the potential underlying mechanism. Stimulation with exogenous IL6 was adequate to mediate lipolysis in adipocytes as indicated by decreased lipid staining (Appendix Fig S2F). Similarly, treatment with IL6-neutralising antibody could reduce the observed lipolysis (Appendix Fig S2G). Mechanistically, IL6 may mediate lipolysis in adipocytes through PKA activation and loss of perilipin (Appendix Fig S2H and I). Perilipin plays a central role in regulating adipose lipolysis and associated rise in serum levels of free fatty acids (Tansey et al, 2004; Zhai et al, 2010). Consistent with this, mice bearing castration-resistant tumours showed significant loss of perilipin in the epididymal adipose tissue along with elevated serum levels of free fatty acids (FFAs), suggestive of host adipose lipolysis (Fig 4K and L). Tocilizumab treatment diminished the observed increase in systemic FFAs (Fig 4M). Importantly, in CRPC patients, serum IL6 levels significantly correlated with serum levels of FFA

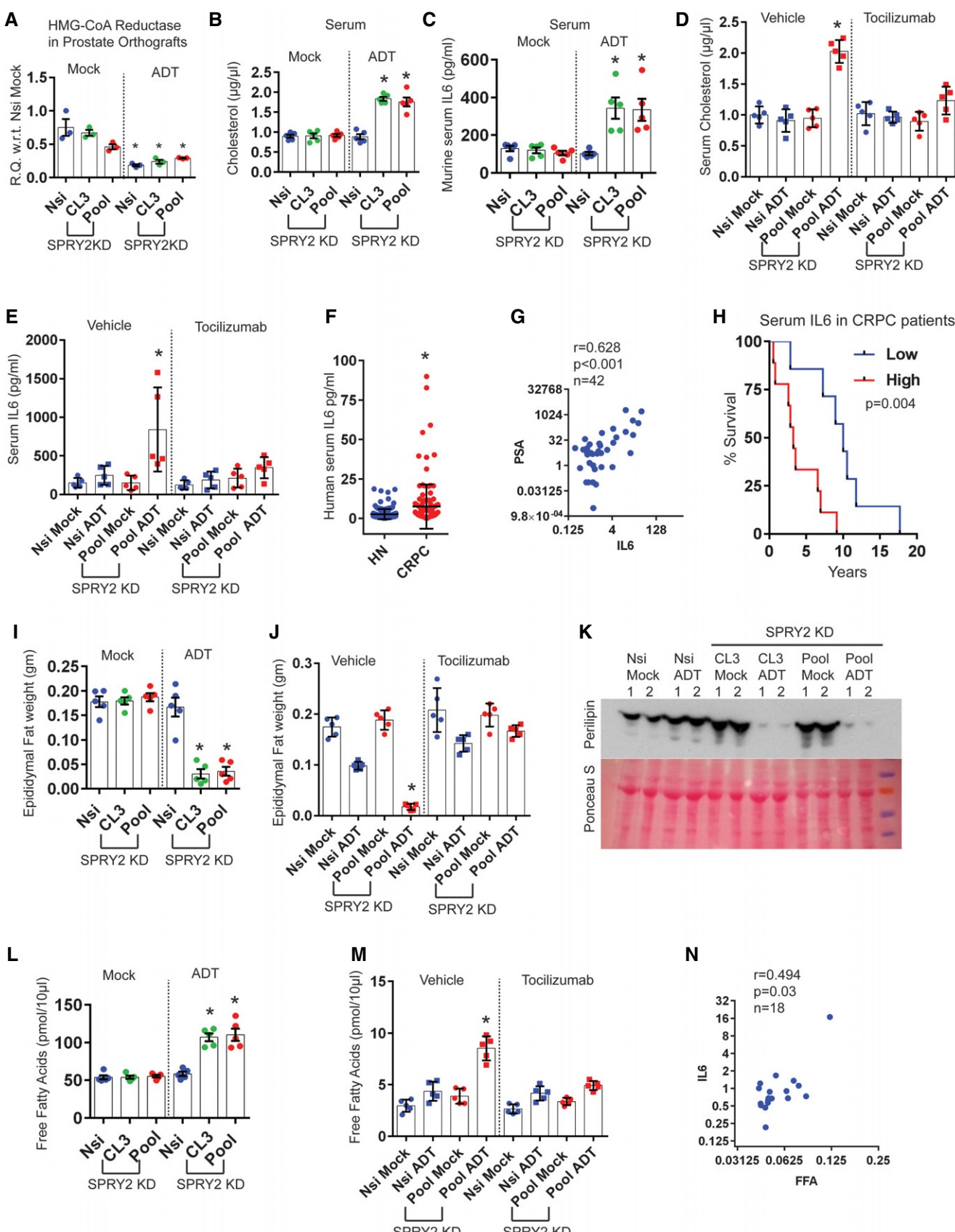

**Figure 4.**

◀

**Figure 4.  Tumour-induced IL6 increases systemic cholesterol levels.**

A        Relative quantitation of human HMGCR mRNA in the indicated CWR22Res orthografts ($n$ = 3 mice per group; *$P$ < 0.05 ANOVA Tukey's test).
B–E    Total cholesterol (B, D) and IL6 (C, E) in sera from mice with CWR22Res prostate orthografts ($n$ = 5 mice per group; *$P$ < 0.05 ANOVA Tukey's test).
F        Serum levels of human IL6 in patients with hormone-naïve ($n$ = 172) or castration-resistant ($n$ = 129) form of prostate cancer (*$P$ = 0.0002; unpaired two-tailed $t$-test with Welch's correction).
G        Scatter plot showing correlation of serum levels of PSA and IL6 from prostate cancer patients.
H        Kaplan–Meier plot for overall (post-diagnosis) survival of CRPC patients with no evidence of metastases (M0) segregated based on above median (high) and below median (low) serum levels of IL6 (high IL6 = 9; low IL6 = 7; log-rank Mantel–Cox test; $P$ = 0.004).
I, J    The epididymal fat weights of mice bearing indicated CWR22Res orthografts ($n$ = 5 mice per group; *$P$ < 0.05 ANOVA Tukey's test).
K        Representative immunoblot image for perilipin in epididymal adipose tissue lysates from mice with CWR22Res prostate orthografts.
L, M    Free fatty acids in sera from mice with indicated CWR22Res prostate orthografts ($n$ = 5 mice per group; *$P$ < 0.05 ANOVA Tukey's test).
N        Scatter plot showing correlation of serum levels of IL6 and free fatty acids (FFAs) from prostate cancer patients.

Data Information: In (A–G, I, J, L–N), each data point represents one independent observation. In (A–E; I, J, L, M), the data are presented as mean ± SD.
Source data are available online for this figure.

(Fig 4N). Thus, systemic IL6 cytokine axis-induced host lipolysis may increase both systemic and tumoral cholesterol levels.

## SPRY2 deficiency-induced IL6 stimulates hepatic cholesterol synthesis and tumoral SRB1 lipoprotein receptor expression

Mice with CRPC have elevated levels of circulating FFAs and cholesterol. Circulating FFAs are typically cleared by the liver, predominantly by esterification to cholesterol. Liver also plays a central role in cholesterol biosynthesis and transport. Hence, we explored the hepatic function in mice bearing CPRC as a potential contributor to elevated levels of circulating cholesterol. Indicating altered liver function, mice with CRPC showed raised serum alanine transaminase activity and increased lipid droplets in the liver (Appendix Fig S2J and K). Interestingly, ADT treatment itself, in non-tumour-bearing mice, stimulated the expression of hepatic HMGCR, which may suggest that elevated hepatic cholesterol biosynthesis may be driven by systemic androgen deprivation (Appendix Fig S2L). Hepatic HMGCR expression is further upregulated in mice bearing CRPC and tocilizumab treatment normalised this effect (Fig 5A). Consistent with the previous report (Flint et al, 2016), our observations support the idea that tumour-induced IL6 may modulate hepatic function.

Overall, our work thus far indicated the potential contribution of hepatic HMGCR function in elevating circulating cholesterol levels, which may then serve as a source for tumoral cholesterol. To evaluate the importance of cholesterol homeostasis in CRPC, we exploited the well-documented effects of statins such as simvastatin (Del Puppo0 et al, 1995) to suppress systemic cholesterol biosynthesis. Treatment with simvastatin significantly sensitised castration-resistant orthografts to ADT (Fig 5B, Appendix Fig S2M and N). Combined treatment of statin with ADT decreased the AR levels in SPRY2-deficient orthografts (Appendix Fig S2O).

To investigate the uptake of cholesterol into the tumours, we evaluated the tumoral levels of lipoprotein receptors that mediate cholesterol transport and tissue distribution. SPRY2-deficient cells showed enrichment of genes involved in lipoprotein binding suggesting an active uptake of lipoproteins and their lipid content (Fig 5C). While the expression of the low-density lipoprotein receptor (LDLR) remained unaltered, CRPC orthografts showed enhanced expression of scavenger receptor B1 (SRB1), a bidirectional high-density lipoprotein (HDL) cholesterol receptor (Fig 5D and E). Importantly, tocilizumab treatment significantly decreased SRB1

expression in CRPC orthografts (Fig 5F). Similarly, p38 inhibition also decreased SRB1 expression in CWR22Res cell (Appendix Fig S3A). We next investigated SRB1 levels in other CRPC models. VCaP CRPC orthografts from castrated mice maintained the nuclear AR levels and showed a dramatic increase in SRB1 staining (Fig 5G, Appendix Fig S3B). Similarly, ADT-resistant NPS tumours also showed increased p-p38 and SRB1 staining (Fig 5H, Appendix Fig S3C).

## SPRY2 deficiency mediates CRPC by increasing tumoral cholesterol uptake through SRB1

We applied two complementary approaches (i) to genetically knock out (KO) SRB1 expression (Fig EV5A and B) and (ii) to chemically suppress SRB1 function with ITX5061 (a specific SRB1 antagonist; Syder et al, 2011) in CWR22Res cells to investigate the functional role of SRB1 in CRPC. Both SRB1 KO and ITX5061 treatment significantly abrogated the growth advantage conferred by SPRY2 deficiency under ADT conditions (Fig 6A). ITX5061 treatment did not exhibit additional growth inhibition in SRB1 KO SPRY2-deficient cells (Fig 6A). These data suggest that (i) SRB1 is required for conferring the growth advantage to SPRY2-deficient cells under ADT conditions, and (ii) ITX5061 mediated majority of its growth inhibitory effects through SRB1 (Figs 6A and EV5C). Mechanistically, SRB1 conferred growth advantage by enhancing HDL uptake under ADT conditions (Fig 6B). SRB1 KO or ITX5061 treatment did not, however, alter cellular LDL uptake (Fig EV5D). Similarly, LNCaP and VCaP cells also uptake more HDL under ADT conditions (Fig EV5E and F). Furthermore, ITX5061 treatment significantly decreased the HDL uptake in LNCaP, VCaP and CWR22RV1 cells (Fig EV5G–I). Since SRB1 is a bidirectional cholesterol transporter, we also tested the effects of ITX5061 on cholesterol efflux. SPRY2-deficient cells effluxed less cholesterol compared to the control cells (Fig EV5J). ITX5061 treatment did not affect the overall cholesterol efflux, suggesting the potential inhibitory role of ITX5061 in HDL cholesterol uptake (Fig EV5J). ITX5061 decreased the growth of LNCaP cells in hormone-deficient conditions (Fig EV5K). ITX5061 further inhibited the growth of CSS-sensitive SPRY2-overexpressing LNCaP cells (Figs 1H and EV5K). ITX5061 treatment also decreased the growth of CWR22RV1 CRPC cells (Fig EV5L).

To investigate the therapeutic importance of targeting SRB1, we treated CWR22Res orthograft-bearing mice with ITX5061. Treatment with ITX5061 significantly sensitised CRPC orthografts to ADT

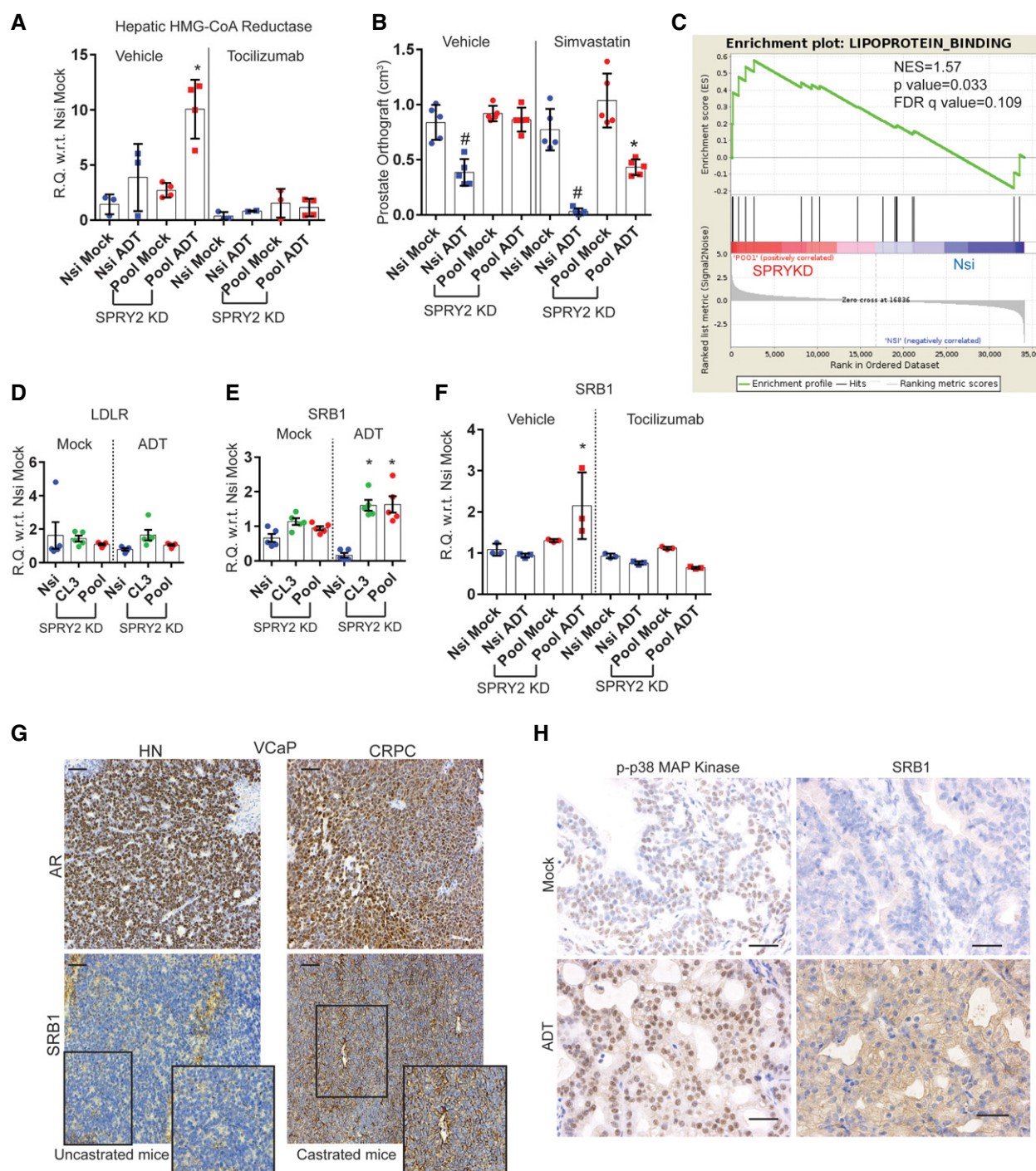

**Figure 5. IL6 stimulates hepatic cholesterol synthesis and SRB1 expression in CRPC.**

A Relative quantitation of murine Hmgcr mRNA in livers from mice with CWR22Res orthografts as indicated ($n = 3$ mice per group; *$P < 0.05$ ANOVA Tukey's test).

B The effects of ADT and simvastatin on prostate orthografts represented as volume ($n = 5$ mice per group; #$P < 0.05$ for Nsi orthografts; *$P < 0.05$ for Pool orthografts; ANOVA Tukey's test).

C GSEA analyses showed enrichment of lipoprotein binding genes in CWR22Res SPRY2-deficient prostate cancer cells.

D–F Relative quantitation of human LDLR (D) and human SRB1 (E, F) mRNA in CWR22Res prostate orthografts ($n = 5$ mice per group for D & E and $n = 3$ mice per group for F; *$P < 0.05$ ANOVA Tukey's test).

G Representative images ($n = 3$) of AR and SRB1 immunoreactivity in VCaP prostate orthograft grown in intact (uncastrated) or castrated mice, signifying VCaP hormone-naïve (HN) and VCaP castration resistance (CR) prostate cancer, respectively. Scale bar = 10 μm.

H Representative images ($n = 3$) of immunostained prostate sections from NPS mice as indicated. Scale bar = 10 μm.

Data Information: In (A, B, D–F), each data point represents one independent observation. In (A, B, D–F), the data are presented as mean ± SD.

(Figs 6C and EV5M and N). Mechanistically, ITX5061 (cholesterol transport antagonist) and simvastatin (inhibitor of cholesterol biosynthesis) treatments significantly decreased testosterone and its precursor, androstenedione within the prostate orthografts under ADT conditions (Fig 6D and E). ITX5061 treated orthografts showed substantially lower levels of AR (Fig EV5O). In NPS mice, ITX5061 treatment significantly induced tumour regression, substantially reducing nuclear AR level (Fig 6F and G, Appendix Fig S3D). Unlike IL6-neutralising therapies, we did not observe any evidence of accelerated formation of visceral metastases in both orthograft and NPS models following ITX5061 treatment (Fig 6H, Appendix Fig S3E). The comparative growth inhibition study conferred higher growth inhibitory potential to ITX5061 treatment compared to simvastatin treatment in SPRY2-deficient CWR22Res and LNCaP cells under hormone-deprived conditions (Appendix Fig S3F). Together, our data indicate that ITX5061 may inhibit CRPC by inhibiting tumoral cholesterol trafficking and simvastatin may mitigate CRPC by regulating systemic cholesterol homeostasis.

To test the significance of SRB1 expression in clinical prostate cancer, we characterised SRB1 expression by immunohistochemistry (Fig 7A). Following ADT, the tumours with high levels of SRB1 progressed to CRPC significantly faster (median time to CRPC = 30 months) than those tumours with lower SRB1 expression (median time to CRPC = 48 months) (Fig 7B). Furthermore, the SR-B1 expression at diagnosis also correlated significantly with overall survival following treatment with ADT (Fig 7C).

Overall, we show that SPRY2 deficiency leads to an androgen self-sufficient form of CRPC. Mechanistically, loss of the tumour suppressor SPRY2 leads to activation of the HER2-IL6 cytokine signalling axis, which enhances tumoral expression of androgen biosynthetic enzyme HSD3B1 and lipoprotein receptor SRB1. At a systemic level, IL6 induces host adipose lipolysis and stimulates hepatic cholesterol biosynthesis leading to increased circulating cholesterol. Thus, SPRY2 deficiency-driven p38-IL6 signalling axis may support resistance to ADT by promoting self-sufficiency for androgens within the tumour, at least in part, through increased levels of intra-tumoral cholesterol required for androgen biosynthesis (Fig 7D). Here, our key conclusion is that restoration of systemic cholesterol homeostasis with the use of statins or the blockade of SRB1-mediated tumoral cholesterol uptake can be potential strategies to diminish treatment resistance in a subgroup of patients at risk of developing CRPC.

## Discussion

Therapeutic regimes designed to impair AR pathway activity remain the first-line treatment for metastatic prostate cancer patients. Most patients, even on the improved androgen-AR-targeted therapies, eventually progress to castration-resistant prostate cancer (CRPC), a lethal stage of the disease. Although certain CRPC forms are AR-independent such as neuroendocrine (NEPC) and recently characterised double-negative prostate cancer (DNPC—AR-negative neuroendocrine negative), almost half of the CRPCs show reactivation of AR pathway (ARPC) (Watson et al, 2015; Bluemn et al, 2017). Defining the mediators of treatment resistance is critical in designing effective therapeutic interventions. Experimental modelling of clinically relevant genomic lesions found in prostate cancer can improve our understanding of treatment resistance (Roychowdhury & Chinnaiyan, 2016). Loss of SPRY2 may represent one such clinically relevant lesion, as patients with SPRY2-deficient cancers show accelerated progression to CRPC with poor overall survival. Here, using both in vitro and in vivo model systems including human prostate orthograft and genetically modified (Gao et al, 2012) pre-clinical murine models, we uncovered a functional role of SPRY2 deficiency in the progression of prostate cancer to CRPC state through bidirectional tumour–host interactions.

We show that SPRY2 loss leads to an androgen self-sufficient form of CRPC representing an ARPC (AR-active prostate cancer) type of clinical CRPC (Watson et al, 2015). We have previously shown how SPRY2 deficiency cooperates with loss of PTEN or PP2A tumour suppressor activity to drive prostate cancer initiation (Patel et al, 2013). We have also previously shown that SPRY2 deficiency in a PTEN-dependent manner may amplify ligand-mediated activation of EGFR/HER2 RTK signalling axis (Gao et al, 2012). While HER2 activation can drive the growth of SPRY2-deficient cells in a PTEN-dependent manner, our current work further reveals that SPRY2 deficiency can mediate progression to an AR-active CRPC state irrespective of PTEN status. Mechanistically, the HER2-p38 signalling axis drives progression of SPRY2-deficient tumours to an androgen autonomous castration-resistant state by inducing IL6 cytokine axis. The IL6 cytokine axis induces tumoral expression of HSD3B1, a key androgen biosynthetic enzyme, and elevate tumoral cholesterol uptake by enhancing the levels of scavenger receptor B1 (SRB1), an HDL receptor. The increase in IL6 levels in CRPC tumours is closely associated with a systemic rise in circulating IL6.

---

**Figure 6. SPRY2 deficiency-induced SRB1 mediates CRPC.**

A    CWR22Res Nsi and SPRY2 (Pool) KD cells were subjected to SRB1 KO and/or treatment with ITX5061 as indicated. Cell numbers were normalised to T0 (Day 0) for each cell line (n = 3; *P < 0.05 ANOVA Tukey's test for all cell lines compared to Pool VC).

B    HDL uptake in indicated cells grown in medium containing 10% FBS (FM) or 10% charcoal-stripped serum (ADT) treated with 15 μM ITX5061 (n = 3; *P < 0.05; [#]P < 0.05 Nsi ADT CTRL compared to Pool ADT CTRL; ANOVA Tukey's test).

C    The effects of ITX5061 on CWR22Res prostate orthografts represented as tumour volume (n = 5 mice per group; [#]P < 0.05 for Nsi orthografts; *P < 0.05 for Pool orthografts; ANOVA Tukey's test).

D, E    LC-MS-based detection of androstenedione (testosterone precursor) ([#]P < 0.05 for Nsi orthografts compared to Pool ADT vehicle; *P < 0.05 for Pool orthografts; ANOVA Tukey's test) (D) and testosterone ([#]P < 0.05 unpaired two-tailed Student's t-test Nsi mock compared to Nsi ADT for respective treatments; *P < 0.05 ANOVA Tukey's test compared to Pool ADT vehicle) (E) in CWR22Res prostate orthografts from mice treated as indicated (n = 3 mice per group).

F    Prostate tumour weights from NPS mice treated with ITX5061 (n = 6; *P < 0.05 ANOVA Tukey's test).

G    Representative images (n = 5) of H&E and immunostained prostate sections from NPS mice treated with ITX5061. Scale bar = 10 μm.

H    Incidence of cumulative visceral metastases was analysed from proximal and distal metastatic sites from mice bearing CWR22Res Nsi or Pool SPRY2 KD orthografts, receiving treatments as indicated. Classification of proximal and distal metastases is shown in Appendix Fig S1A.

Data Information: In (C–F), each data point represents one independent observation. In (A), each data point represents mean ± SD. In (H), data presented as contingency graph. In (B–F), the data are presented as mean ± SD.

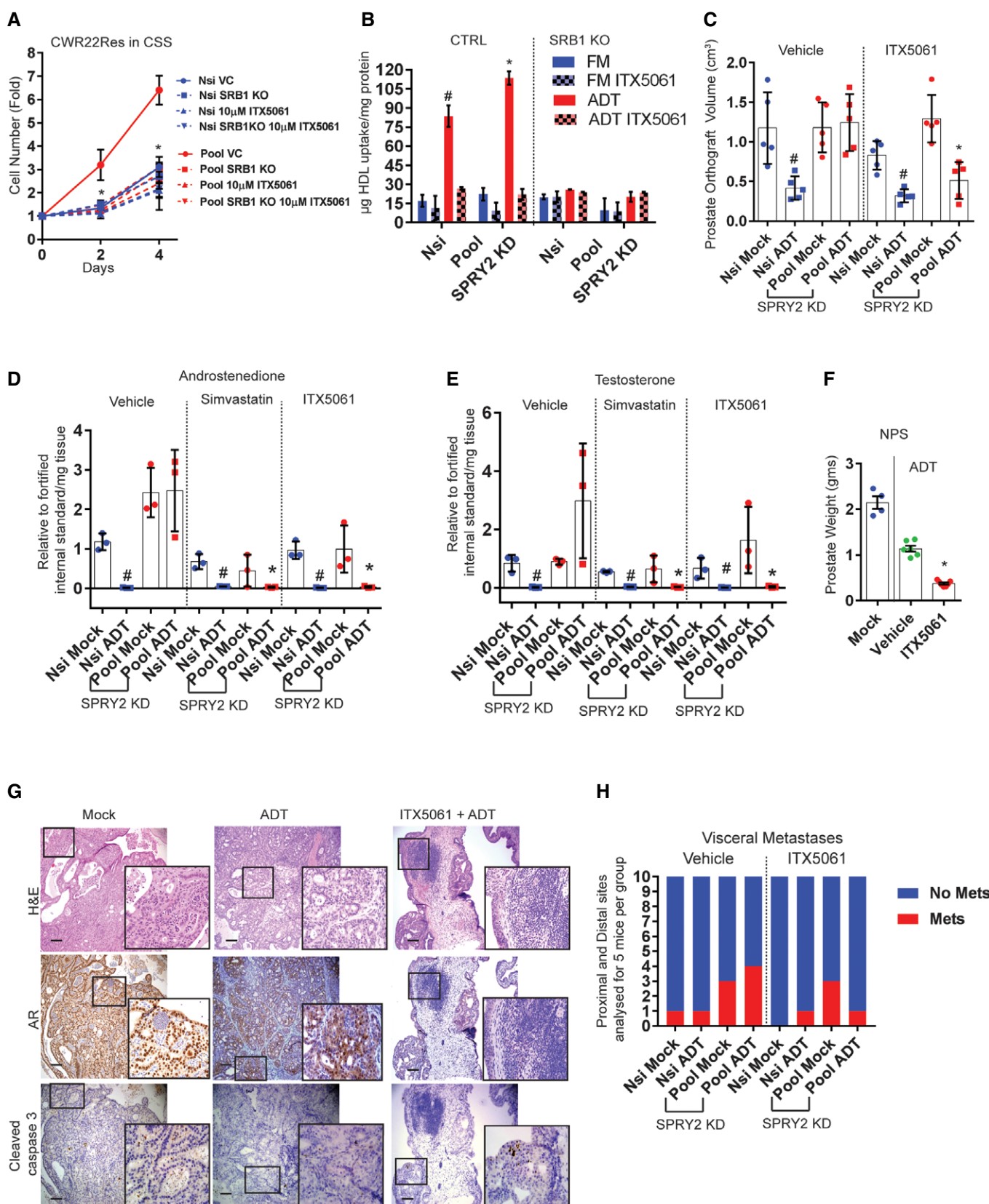

**Figure 6.**

Furthermore, IL6-mediated systemic effects such as host lipolysis and altered hepatic cholesterol homeostasis may aid disease progression to CRPC through elevated levels of circulating cholesterol. Thus, IL6 at both tumoral and systemic levels may drive disease progression.

Besides full-length AR, both *in vitro*-cultured CWR22Res cells and the *in vivo* orthograft model used in this study also express AR variants that can promote castration resistance (Watson *et al*, 2015). Although AR variants may bypass the requirement of ligand due to the absence of ligand binding domain, previous studies suggest that AR variants including ARV7 may still respond to anti-AR treatments targeting the ligand binding and continue to require full-length AR to mitigate their function (Watson *et al*, 2010). In our model system, treatments that sensitised orthograft response to ADT also suppressed the expression of full-length and variant AR. Future characterisation of the AR transcriptional activity under the conditions that sensitised SPRY2-deficient models to ADT may provide further understanding of AR reactivation in CRPC.

Our work highlights the importance of HER2 signalling in mediating progression of SPRY2-deficient cancer to CRPC. HER2 and the downstream signalling cascades are potentially clinically actionable targets and may play a functional role in driving treatment resistance (Robinson *et al*, 2015; Shiota *et al*, 2015; Roychowdhury & Chinnaiyan, 2016). In early phase I and II clinical trials, while HER/EGFR inhibitors do not alter the disease progression in untreated hormone-naïve patients, a subset of CRPC patients treated with lapatinib, a HER2/EGFR inhibitor, did show a decrease in disease progression (Sridhar *et al*, 2010; Whang *et al*, 2013). Likewise, our work underscores the importance of HER2 in promoting tumour growth only under ADT condition. Our work implicates a central role of IL6 in HER2-mediated CRPC in SPRY2-deficient tumours. IL6 cytokine signalling axis has been implicated in prostate cancer progression and treatment resistance (Malinowska *et al*, 2009; Culig, 2014). IL6 being a pleiotropic cytokine may arise from multiple sources in cancer patients. While reports indicate little IL6 production from primary or metastatic clinical prostate cancer samples, the IL6 receptor is present uniformly in a majority of prostate cancers (Siegall *et al*, 1990; Culig, 2014; Yu *et al*, 2015). Importance of molecular events such as SPRY2 loss or HER2 activation in tumoral IL6 production in clinical prostate cancers needs further investigation. The tumour microenvironment including infiltrating immune cells and stromal tissue including prostate cancer-associated fibroblasts may also substantially contribute to tumoral IL6 (Doldi *et al*, 2015). IL6 has been shown to promote both androgen-dependent and androgen-independent forms of CPRP either by inducing tumoral androgen biosynthetic cascade or by modulating ligand-independent functions of AR (Kim *et al*, 2004; Chun *et al*, 2009). Our work suggests that in SPRY2-deficient tumours IL6 may mediate AR-dependent CRPC formation by facilitating tumoral androgen biosynthesis, at least in part through enhanced androgen biosynthesis. Amongst the androgen biosynthetic enzymes, while the expression of CYP17A1 may not consistently alter based on SPRY2 status, the HSD3B1 levels were elevated in all SPRY2-deficient CRPC models studied. In addition to the expression of CYP17A1, HSD3B1 and HSD17B1, analyses of their enzymatic activities in SPRY2-deficient cells may further aid in ascertaining the functional contributions of these androgen biosynthetic enzymes in CRPC progression.

Accumulation of HSD3B1 protein is found to be sufficient to facilitate the emergence of CRPC by stimulating testosterone biosynthesis from precursor sterols (Chang *et al*, 2013). Recently, the HSD3B1 status has been validated in prostate cancer patients as a robust molecular determinant to predict unfavourable response to ADT through testosterone synthesis (Hearn *et al*, 2016). Sharifi and colleagues also implicate the potential role of HSD3B1 in developing resistant to the CYP17A1 inhibitor abiraterone (Chang *et al*, 2013). Abiraterone has significantly improved the prognosis and overall survival in patients with advanced CRPC (Sartor & Pal, 2013). We find that combining abiraterone with HSD3B1 depletion maximised the growth inhibition of SPRY2-deficient CWR22Res cells in hormone-depleted conditions. Of note, SPRY2 status did not alter the response to abiraterone treatment in LNCaP cells. Both SPRY2-proficient and SPRY2-deficient LNCaP cells exhibited similar abiraterone-induced growth inhibition. Whether the differences observed between CWR22Res and LNCaP models are a result of PTEN status or presence of AR variants needs further investigation. Sharifi and colleagues have recently shown that D4A, an active abiraterone metabolite, can inhibit multiple androgen biosynthetic enzymes and AR (Li *et al*, 2015). Whether D4A, a pan inhibitor of androgen biosynthetic enzymes, sensitised CRPC with high HSD3B1 expression (e.g., SPRY2-deficient CRPC tumours) to ADT requires further investigation.

In addition to steroid biosynthetic enzymes, cholesterol is an important precursor for androgen biosynthesis, even in advanced prostate cancer (Dillard *et al*, 2008). Majority of the cholesterol is transported to various organs within low (LDL)- and high (HDL)-density lipoproteins (Hu *et al*, 2010). Both LDLR (LDL receptor) and SRB1 (HDL receptor) have been implicated in tumoral cholesterol uptake and disease progression (Leon *et al*, 2010). While tumoral LDLR levels did not change, we observed enhanced SRB1 expression and HDL uptake in SPRY2-deficient ADT-treated orthografts. Although SRB1 is a bidirectional cholesterol transporter, SPRY2-deficient tumour cells take up more and efflux less cholesterol. This may

**Figure 7.  SRB1 in clinical prostate cancer.**

A   Representative SRB1 immunostained images of prostate cancer samples. High SRB1 (*n* = 47) represents patients with SR-B1 IHC score above median, and low SRB1 (*n* = 43) represents patients with SR-B1 IHC score below median (scale bar = 10 μm).

B   Kaplan–Meier plot for relapse-free survival of ADT-treated prostate cancer patients (SRB1-high patients = 11; SRB1-low patients = 11; log-rank Mantel–Cox test; *P* = 0.006).

C   Kaplan–Meier plot for overall (post-diagnosis) survival of patients treated with ADT (SRB1-high patients, *n* = 12; SRB1-low patients, *n* = 16; log-rank Mantel–Cox test; *P* = 0.0354).

D   SPRY2 deficiency facilitates progression of prostate cancer to CRPC through HER2-mediated induction of IL6, HSD3B1 and SRB1. IL6 cytokine axis, in a paracrine manner, induces host adipose lipolysis and hepatic cholesterol synthesis, resulting in increased circulating cholesterol. ADT-resistant tumours take up cholesterol through SRB1 for androgen biosynthesis. Normalising the systemic cholesterol homeostasis by statins and blocking SRB1-mediated cholesterol uptake by tumours may serve as potential approaches to diminish treatment resistance in a subset of prostate cancers with the SPRY2 deficiency or HER2 activation.

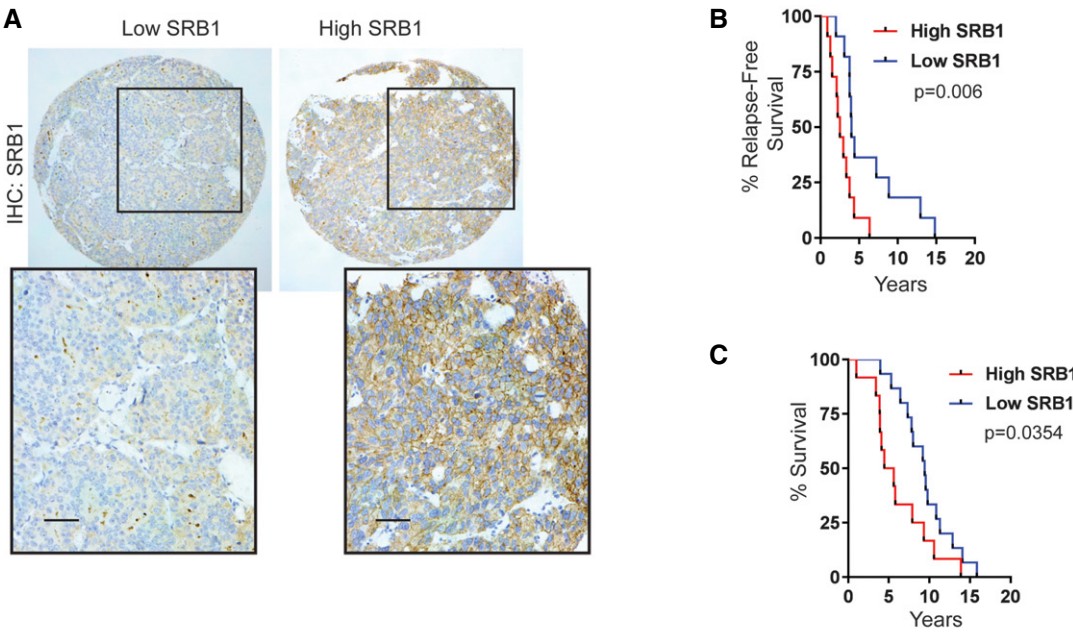

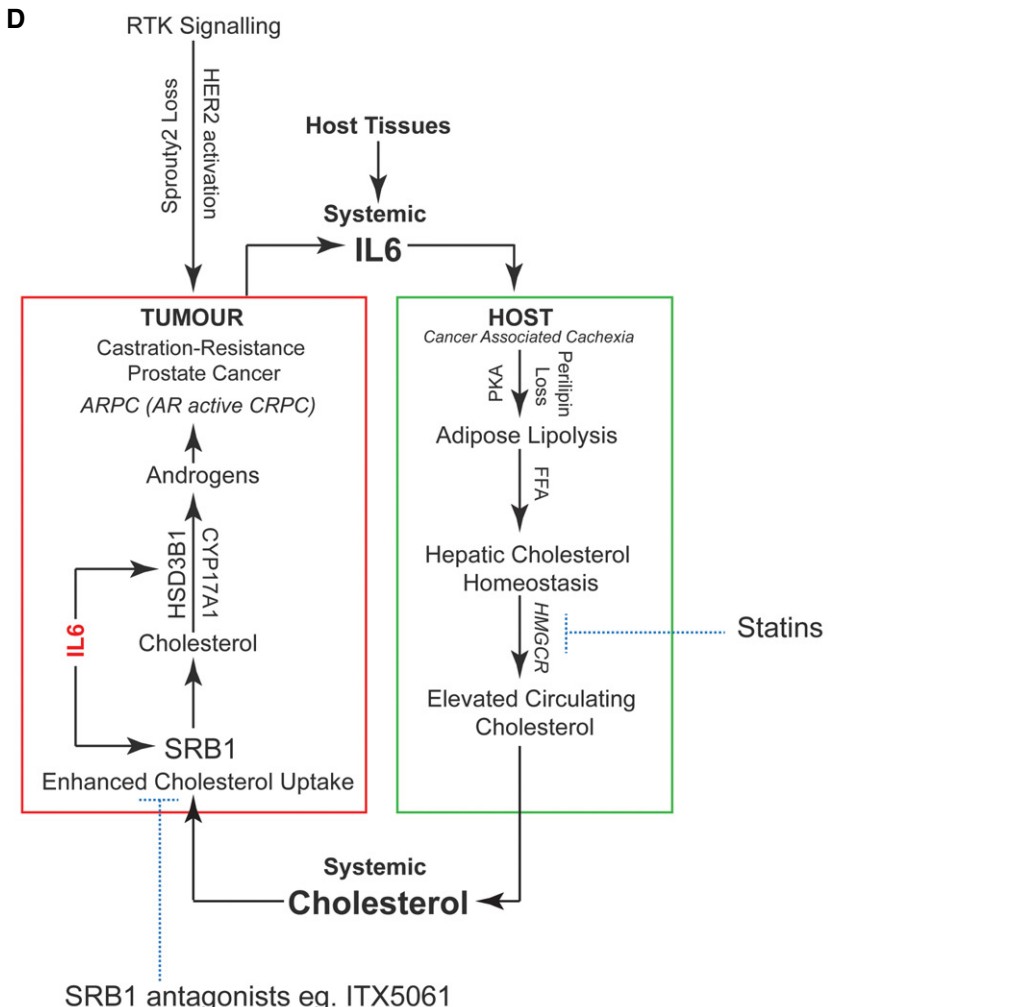

**Figure 7.**

suggest cellular requirement to retain and use the cholesterol for biomass and other biosynthetic needs including steroid biosynthesis. Similar to our findings, SRB1 has a pro-survival function in C4-2 cells, which is a CRPC variant of LNCaP cells (Twiddy et al, 2012). Furthermore, we find that cholesterol uptake by SRB1 seems to be essential for the growth and survival of SPRY2-deficient cells under ADT conditions. To date, the mechanisms that regulate SRB1 expression in CRPC have not been characterised. Here, we show that besides regulating HSD3B1 expression, IL6 also regulates SRB1-mediated cholesterol uptake in SPRY2-deficient CRPC cells. Consistent with our clinical findings, a recent analysis of lipoprotein receptors in clinical prostate cancer showed that SRB1 mRNA correlated with HSD3B1 expression, and patients with high tumoral SRB1 levels had shorter progression-free survival (Schorghofer et al, 2015).

The value of targeting the IL6 pathway in prostate cancer including CRPC remains unclear (Culig, 2014). While IL6-targeting therapies have not demonstrated clear effects on clinical outcomes such as progression-free survival, there appeared to be a subgroup of patients that demonstrated good PSA response as well as clinical and radiologic evidence of stable disease (Dorff et al, 2010; Fizazi et al, 2012). In our two complementary in vivo models, IL6-neutralising therapies paradoxically increased the incidence of metastasis despite sensitisation of primary tumours of ADT. Similar counterintuitive increase in metastatic burden following anti-IL6 treatment was recently reported in Pten-deficient prostate cancer model (Pencik et al, 2015). The potential explanation for such unexpected pro-metastasis effects following IL6-targeted treatment may be due to the abrogation of IL6-STAT3 signalling-mediated regulation of tumour suppressors such as TP53 (Pencik et al, 2015). Collectively, our in vivo data on IL6-neutralising treatment suggest a complex pattern of response, with improved local tumour response and increased metastatic incidence. Despite evidence of increased metastases in animals treated with IL6-neutralising therapies, we did not find any correlation between serum IL6 levels and overall metastatic burden in prostate cancer patients. This lack of association may be attributed to a number of factors including the pleiotropic nature of IL6, multiple IL6 sources and systemic inflammation that is frequently associated with widespread metastases.

By understanding the disease progression at an organism level, we find that IL6, in addition to mediating the tumoral treatment resistance, also plays a central role in governing tumour–host interactions to promote disease progression. As previously summarised (Culig, 2014), we find that CRPC patients with high serum IL6 have poor survival. Although systemic IL6 is closely associated with poor patient survival, its functional role in facilitating CRPC by influencing host metabolism has not been fully understood. Tumour-induced IL6 has been previously shown to mediate cachexia and treatment response in other cancer models (Flint et al, 2016). Increase in circulating IL6 can be a consequence of tumour-induced inflammation. Here, we report that in addition to tumoral IL6 production, host visceral adipose tissue may, at least in part, contribute to the rise in systemic IL6. This elevated circulating IL6 may, in turn, induce host adipose lipolysis (Rossi et al, 2015). In both pre-clinical treatment-resistant mouse models and clinical CRPC, we found elevated circulating levels of IL6 and free fatty acids (FFAs). IL6 mediates lipolysis in adipocytes by inducing the loss of lipid droplet-associated protein perilipin, which protects lipids within the droplet from intracellular lipases. IL6-mediated inflammation can also alter hepatic cholesterol

homeostasis by inducing hepatic cholesterol biosynthesis (Zhao et al, 2011). Consistent with this, systemic IL6 also stimulates hepatic cholesterol biosynthesis in mice with CRPC leading to dyslipidaemia with elevated levels of circulating cholesterol. Thus, at a systemic level, IL6 rewires host metabolism to raise circulating cholesterol levels that may further facilitate prostate cancer progression towards a CRPC phenotype. Highlighting the importance of cholesterol in the development of treatment resistance and overall prognosis, retrospective clinical investigations found that men who used statins after diagnosis of prostate cancer had a 24% decreased risk of cancer-associated mortality (Yu et al, 2014). Similar to the reported effectiveness of statins in inhibiting the progression of LNCaP tumours to CRPC state (Wang et al, 2014), we show that decreasing cholesterol bioavailability using statin treatment sensitised SPRY2-deficient tumours to ADT. ITX5061 was previously developed as an anti-viral agent (Syder et al, 2011) and was found to be a clinically safe SRB1 specific antagonist. Our data support the repurposing of ITX5061 in treating CRPC. ITX5061 treatment re-sensitised prostate tumours to ADT by inhibiting SRB1-mediated cholesterol uptake, thereby decreasing tumour androgen biosynthesis. Further investigations are therefore warranted to robustly examine the effectiveness and safety of SRB1 antagonist such as ITX5061 to tackle treatment-resistant prostate cancer. Additional pre-clinical models will be required to identify suitable patient groups and to define the optimal treatment schedule in using ITX5061.

In our model, statins and ITX5061 act at two distinct nodes of disease progression to sensitise tumours to ADT. While statins may act by normalising hepatic cholesterol biosynthesis, ITX5061 as an SRB1 antagonist may work by decreasing tumoral cholesterol uptake. By studying CRPC at both tumour intrinsic (intra-tumoral) and extrinsic (host) levels, we have identified two actionable nodes that can sensitise prostate tumours to ADT. Thus, prostate patients with SPRY2 loss or HER2 activation may progress to form androgen autonomous AR-activated CRPC. This type of treatment-resistant cancer may respond to treatments targeting systemic cholesterol bioavailability (statins) and tumoral cholesterol transport (SRB1 antagonist, ITX5061).

# Materials and Methods

### Study approvals

All the animal experiments conducted for this study were carried out with ethical approval from University of Glasgow under the revised Animal (Scientific Procedures) Act 1986 and the EU Directive 2010/63/EU (PPL 30/3185). For clinical samples, ethical approval was gained from the West of Scotland Research Ethics Committed (05/S0704/94). The informed consent was obtained from all subjects and that the experiments conformed to the principles set out in the WMA Declaration of Helsinki and the Department of Health and Human Services Belmont Report.

### Murine prostate cancer models

For prostate orthograft animal experiments, $14 \times 10^6$ CWR22Res cells (Appendix Table S1) were surgically implanted in one of the anterior prostate lobes of CD1-nude male mice (6–8 weeks old).

CWR22Res prostate cancer cells with stable knockdown of SPRY2 were generated as described in Appendix information. After 30 days when a palpable tumour was felt with 100% tumour incidence, the mice were randomised into two groups: mock (sham surgical incision) and ADT (orchiectomy). A refined 60-days timed protocol was used for all the treatment experiments. *Nkx 3.1-Cre* mice were crossed to those harbouring *Spry2*$^{fl/+}$ and *Pten*$^{fl/+}$, and mice were genotyped by PCR by Transnetyx™. After approximately 50 weeks, the *Nkx 3.1-Cre Pten*$^{fl/+}$ *Spry2*$^{fl/+}$ (NPS) developed palpable prostate tumours (Gao *et al*, 2012). The mice with palpable prostate tumours were randomised into two groups—mock (sham surgical incision) and androgen deprivation therapy (ADT), which was achieved by orchiectomy. NPS mice with mock or ADT treatment were sacrificed 1 month after treatment, and a subset of these mice was aged to clinical end point. Drug treatments were initiated 3 days post-castration (ADT): tocilizumab (100 μg in PBS given as I.P. injection 3 times a week for 3 weeks), simvastatin (80 mg/kg/day in 30% PEG400 + 0.5% Tween 80 + 5% propylene glycol in water, gavaged daily for 1 month), and ITX5061 (25 mg/kg/day in 20% hydroxypropyl-beta-cyclodextrin made in 20 mM citric acid, gavaged once daily for 1 month). Proximal metastases are defined as disease in lumber lymph nodes and epididymal fat pads; distal metastasis is defined as involvement of thoracic lymph nodes (diaphragmatic) and in the lungs. Metastatic deposits were detected by H&E and confirmed by AR immunostaining. Treatment response in prostate cancer orthografts was calculated based on orthograft volume or using Leica image analyser software to measure the total tumour area and central necrotic area. The data were represented as % necrosis/tumour weight.

## Steroid measurements

Serum and intra-tumoral testosterone levels were measured using Testosterone EIA kit (Cayman Chemicals, 582701). For LC-MS-based steroid detections (details provided in supplementary information), the relative peak ratio was obtained by normalising the biological sample peak area to the peak area obtained from the corresponding fortified internal standard.

## Clinical datasets for gene alterations and survival analyses

cBioPortal platform was used to query TCGA provisional prostate cancer dataset. The data was extracted, and Kaplan–Meier survival curves were plotted using GraphPad Prism.

## Real-time PCR

10 μg of RNA was used for cDNA synthesis using high-capacity cDNA reverse transcription kit (Applied Biosystems). Real-time PCR was carried out using TaqMan® Gene Expression Master Mix (Life Technologies) in 7500 Fast Real-Time PCR System (Applied Biosystems). CACS3 was used as the housekeeping gene, and the relative quantities for each assay were obtained by normalising to one biological control sample (as indicated in the figures).

## Studies using clinical serum samples

Serum samples from 172 patients with HNPC and 129 patients with CRPC were obtained from ProMPT study (ethics committee

### The paper explained

#### Problem
Prostate cancer is one of the most common cancers amongst men. Treatment resistance and cancer metastases are leading causes of cancer-associated mortalities in patients with prostate cancer. Prostate cancer patients treated with androgen deprivation therapy (ADT), a standard-of-care treatment for advanced disease, will eventually progress to a lethal treatment-resistant state, referred to as castration-resistant prostate cancer (CRPC). Almost half of CRPC tumours maintain their tumoral androgens and show evidence of androgen receptor (AR) reactivation. Both tumour intrinsic molecular events and host-tumour interactions can govern treatment response. Dual understanding of the tumoral molecular events and tumour–host cross talks in mediating treatment resistance may help identify treatment strategies that may improve the efficacy of current treatment modalities.

#### Results
Using an integrated approach to study treatment resistance in clinical and murine pre-clinical models of prostate cancer, we show that loss of tumour suppressor SPRY2, a negative regulator of receptor tyrosine kinase signalling, leads to CRPC. Mechanistically, SPRY2 deficiency mediates treatment resistance through HER2-driven IL6 cytokine axis. SPRY2 deficiency-induced HER2-IL6 signalling mediates an androgen autonomous form of CRPC through enhanced tumoral (i) HSD3B1 expression, a key androgen biosynthetic enzyme, and (ii) cholesterol via the HDL receptor, SRB1. At a systemic level, the tumour-induced IL6 cytokine axis elevated circulating cholesterol levels by inducing host adipose lipolysis and hepatic cholesterol biosynthesis. Collectively, our data illustrate how tumour-induced IL6 rewires host lipid metabolism to promote treatment resistance by fuelling tumoral androgen biosynthesis through enhanced cholesterol transport.

#### Impact
The tumoral cholesterol transport and systemic cholesterol homeostasis play a significant role in facilitating treatment resistance. We propose targeting cholesterol bioavailability using statins or tumoral cholesterol uptake using ITX5061, a clinically safe SRB1 antagonist, as two strategies to sensitise prostate tumours with SPRY2 deficiency or HER2 activation to the standard-of-care androgen deprivation therapy. Overall, we have identified SPRY2 deficiency as a driver of treatment resistance, systemic IL6 as a circulating prognostic marker and SRB1 as a clinically actionable target.

approval: UK MREC number 01/4/61). Serum levels of PSA, FFA and IL6 levels were measured in the same serum samples from individual patients. The patients' records were used to acquire the data on PSA levels. The study criteria for IL6 and PSA correlation analyses were to obtain the PSA level data on the same serum samples (matched for sample collection date) from individual patients; 42 patients matched these criteria. Hence, IL6 and PSA correlation analyses was performed in 42 samples. For FFA and IL6 correlation analyses, a subset of 20 serum samples was requested from the ProMPT study, randomly selecting 10 HNPC and 10 CRPC cases for which IL6 levels were previously assayed. Out of these 20, 18 samples were analysed for FFA levels. Hence, the FFA and IL6 correlation data presented are for 18 samples.

## Data analyses

Data plotting and statistical analysis were done using Prism Graph Pad 7 (Appendix Table S2).

## Data availability

The Illumina microarray data are deposited in Geo Gene Expression Omnibus: GEO Submission (GSE108456).

**Expanded View** for this article is available online.

## Acknowledgements

This work was funded by Cancer Research UK (grant numbers A15151, A10419 and A17196) and Prostate Cancer UK (grant number PG10-10). We thank the biological services unit for technical and histology services. We thank iTherX for providing ITX5061. We are grateful to Peter Adams, Karen Vousden, Fabio Zani, Dilys Freeman and Flossie Wong-Staal for helpful discussions.

## Author contributions

Conceptualisation, RP, HYL and OJS; mCRPC model development, RP; Methodology, RP, JF, EM, VH, PR, AB, KT, NB and GM; Formal Analyses, RP, JF and CL; Investigations, RP, JF, CL, IA, AB, AH and PR; Resources, RP, JE, FCH, MS and OJS; Data Curation, RP, JF, PR, AB and AH; Writing, RP and HYL; Visualisation, RP; Supervision, RP and HYL; Project Administration, RP and HYL.

## Conflict of interest

The authors declare that they have no conflict of interest.

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
