## [Review Process File · EMBO Molecular Medicine]

Sprouty2 loss induced IL6 drives castration-resistant prostate cancer through scavenger receptor B1

Rachana Patel, Janis Fleming, Ernest Mui, Carolyn Loveridge, Peter Repiscak, Arnaud Blomme, Victoria Harle, Mark Salji, Imran Ahmad, Katy Teo, Freddie C. Hamdy, Ann Hedley, Niels van den Broek, Gillian Mackay, Joanne Edwards, Owen J. Sansom and Hing Y. Leung

Review timeline:

Submission date:	4 August 2017
Editorial Decision:	6 September 2017
Revision received:	3 January 2018
Editorial Decision:	26 January 2018
Revision received:	9 February 2018
Accepted:	20 February 2018

Editors: Roberto Buccione and Céline Carret

Transaction Report:

1st Editorial Decision

6 September 2017

Thank you for the submission of your manuscript to EMBO Molecular Medicine.

I apologise for the delay in getting back to you on your manuscript. In fact, we experienced significant difficulties in securing expert and willing reviewers, and then obtaining their evaluations in a timely fashion, mostly due to the overlapping holiday season. Furthermore, additional internal discussion was required to reach final decision.

You will see that the reviewers raise many serious concerns. These encompass various aspects including lack of clarity and focus, missing controls, poor description of experimental details and somewhat overstated conclusions. More specifically, while reviewer 2 is generally more positive, 1 and 3 are more reserved and suggest that significant experimentation, including *in vivo*, may be required to bring the manuscript to the appropriate level of conclusiveness.

This latter issue and the others were discussed during our cross-commenting exercise and as mentioned above, internally. The consensus that emerged was that although the inclusion of data from additional *in vivo* models would be valuable, this would require a major undertaking and may be possibly unnecessary. Ultimately, the greatest priorities remain to greatly improve the presentation and discussion of the data, to provide an adequate introduction and discussion of the literature and, most importantly, to perform additional experiments to strengthen and substantiate the conclusions as needed to respond to the extensive comments of all three reviewers. Furthermore, the claims relating to the clinical importance of the findings would need to be considerably toned down.

In conclusion, while publication of the paper cannot be considered at this stage, we are willing to consider a substantially revised manuscript, addressing the reviewers' concerns as mentioned above including with further experimentation where required. I will not, however, be asking you to provide more data using additional *in vivo* models, although I would encourage you to include such data, if available. It remains important nevertheless, that you discuss the limitations of your study in this

respect.

I look forward to receiving your revised manuscript in due time.

***** Reviewer's comments *****

Referee #1 (Comments on Novelty/Model System for Author):

Please see remarks to the author regarding the model system.

Referee #1 (Remarks for Author):

The manuscript by Patel et al. describes a variety of downstream effects of Sprouty2 (SPRY2) depletion in a mouse model of prostate cancer (PC) engineered for prostate-specific heterozygous loss of Spry2 and Pten as well as in PC orthotopic xenografts and PC cell-based experiments. This study extends previous work from Leung and collaborators (Gao et al EMBO Mol Med 2012), which demonstrated that heterozygous Spry2 and Pten deficiency results in constitutive EGFR/HER2 signaling. These previous studies further showed that Spry2 deficiency sensitizes PC cells to EGF stimulation in a PTEN-dependent fashion. The current manuscript describes how SPRY2 depletion leads to: 1. HER2-mediated PC cell synthesis of androgens and evasion of standard androgen deprivation therapy (ADT), 2. Induction of IL6 and cholesterol metabolism as well as the ramifications of anti-IL6 therapeutic approaches for primary PC vs. metastases and 3. Increased cholesterol scavenger receptor B1 (SR-B1), which increases cholesterol uptake and may serve as a potential therapeutic target in SPRY2 deficient PC tumors.

While most of the experiments taken individually are carefully conducted and the results sound, the manuscript is very unfocused and lacks some important experimental details. Further, the manuscript lacks a logical flow and is not cohesive making it not only difficult to follow but also hard to evaluate the major conclusions.

Specific Comments:

1. The title is extremely misleading in that it is relevant to only approximately half of the manuscript. Similarly, the introduction is incomplete as it does not mention some of the pathways that are being investigated.
2. There is minimal description of the CWR22 cell line used to generate cells depleted of SPRY2. CWR22 is typically propagated serially as mouse xenografts and it is unclear if the authors used the cell line, CWR22Pc, which is derived from CWR22. Also, the pten status should be stated, which is presumably wild type.
3. The manuscript uses two models with differing pten status. Given that this lab previously identified an important contribution of PTEN to the response of PC in the setting of SPRY2 deficiency, this issue requires examination. Furthermore, the manuscript only includes the Spry2 expression data from patients samples; Pten expression data should also be included.
4. IHC analyses throughout the manuscript require some type of scoring or quantification.
5. Although there is significant focus on IL-6, there is little discussion of how the authors' findings fit into the extensive literature on IL-6 in prostate cancer.
6. There is a differential effect of ADT in the NPS GEMM mouse PC model at 1 month vs 6 months ADT but there is no discussion of the timing of ADT in the orthotopic model nor how the two models compare.
7. It is not clear whether intra-tumoral testosterone levels were quantified by ELISA or LC-MS.
8. The previous paper (Gao et al.) examined the effects of SPRY2 loss in androgen receptor negative PC cells but the current work focuses on an androgen receptor (and possibly androgen receptor splice variant) expressing model yet there is essentially no analysis of the contribution of androgen receptors.
9. The pooled CWR22 vs CL3 cells exhibit different extent of SPRY2 depletion. How might this difference contribute to the regulation of CYP17A1 that is only found in the pooled SPRY2-knock down cells? Similarly, overexpression of SPRY2 in LNCaP cells results in increased CYP17A1 and HSD3B1. These results suggest that the focus solely on HSD3B1 may not be justified.
10. Experiment in figure 3F lacks intact mice as controls.
11. The manuscript would be strengthened by stating the objectives of the study more clearly, improving the logical flow of data presentation and removing extraneous data.

12. It appears that ITX is no better than statins with respect to androstenedione or testosterone levels as compared under ADT conditions. Thus, it is not clear why the authors conclude that SR-B1 represents an exciting new target for PC. The authors should address this issue experimentally.
13. Analysis of the effects of abiraterone should be conducted in at least one of the two SPRY2-deficient model systems.

Referee #2 (Comments on Novelty/Model System for Author):

The authors have used multiple mouse models, including CRISPR knockout and shRNA knockdown cell line in vivo models making it a very comprehensive and quality technical analysis of the topic. While each individual component is not necessarily novel (see comments on related publications that require citing and discussion), the data convincingly link these signalling pathways together, which makes it a novel study with novel findings. Considering the medical implications of repurposing currently approved drugs such as statins or the SR-B1 inhibitor, this makes the medical impact high.

Referee #2 (Remarks for Author):

While the authors have undertaken a comprehensive experimental analysis of the links between RTK signalling, SPRY2, IL6 and cholesterol uptake through SRB1 in prostate cancer, they have neglected to appropriately reference and discuss a number of key publications in SPRY2 and PCa papers (including one of their own PMID: 26075267, PMID: 23150596 and PMID: 23584380), IL-6, statins and PCa (PMID: 24296978) and SR-B1 in PCa (PMID: 19866465, PMID: 18782595 and PMID: 22025344). While these do not detract from the experimental data presented, these should be included and discussed as important precursors for their work - which nicely links these previous findings together.

The authors have done a good job of profiling the effects of SPRY2 deficiency on response to ADT, both in a GEMM het SPRY2/PTEN model, as well as in a cell line with SPRY2 KD. These data convincingly show that SPRY2 loss leads to resistance to ADT. Addition of PTEN status to Figure S2C would be helpful, since the GEMM mouse model combines PTEN and SPRY2 heterozygosity. This is particularly important as the authors have published that CWR22 cells have intact PTEN - and indeed that SPRY2KD does not substantially alter PTEN expression in this same CL3 line. This should be explicitly stated and discussed in the text, as it is a clear difference between the GEMM and the orthograft model.

Since the authors have a good antibody for HSD3B1, it would have been more convincing to perform western blots to confirm qPCR expression in FigS3B, S3D and Fig2B-C and 5I.

The authors have nicely shown that HER2 KD can block IL6 expression. Since "SPRY2 deficiency may confer androgen self-sufficiency by inducing HER2 mediated IL6 cytokine axis", correlating changes in HER2 (Fig S3F) and IL6 (Fig 2L) would assist in supporting these data.

An interesting finding is that anti-IL6 reduces tumour burden, but increases metastasis. Do the CRPC patients with increased serum IL-6 have reduced metastasis (from Figure 4B)?

Why is Figure 4E 42 patients and Figure 4F only 18 patients?

The effects of IL6 on lipolysis and serum lipid species is nicely shown.

Figures S8C and D need some statistical analysis. More explanation of the IC50 analysis undertaken in S8F should be provided. It is hard to interpret the data currently supplied as the range of doses used is not wide enough and there are no details on what is the control in each case (compared to DMSO with parent or KO cell line, FM or CSS?). It is hard to determine how the IC50 data were generated from these curves - and indeed there are no IC50 data for CWR cells.

The Main text, figures and Supplementary data have many mistakes, and should be thoroughly edited for English, spelling, and typographical errors. For example, spelling of Sprouty2 is incorrect on line 57 page 6, "self-sufficient from of CRPC" on line 304 page 13, and there are lots of other

inconsistencies throughout the main figures (e.g. SRPY2 in Figure 6C and D; use of "u" instead of micro in multiple figures etc) and in the supplementary figures (RMPI instead of RPMI, Figure S8 legend says Figure S7, ug and ul instead of micro symbol, consistency of labelling etc).

There is no Figure 7F despite being referred to in the text "Furthermore, the SR-B1 expression at diagnosis also correlated significantly with overall survival in ADT treated patients (Fig. 7F)."

Figure S9C may be more helpful in the main text, as this assists in bringing together the findings in the paper.

Referee #3 (Comments on Novelty/Model System for Author):

The majority of the human xenograft data was generated using only the CWR22 line. Additional lines, particularly the LNCaP should be similarly tested since it has low SPRY2 expression coupled with significant expression of HER2.

Referee #3 (Remarks for Author):

There are several issues with this manuscript, the most significant is whether the results are generalizable to the clinic. The authors state on page 15 that: "Despite the overwhelming clinical occurrence and functional contributions of HER2/RTK signaling in driving treatment resistance in prostate cancer, inhibition targeting RTK's ... have offered limited success." One can argue about the statement about "overwhelming contribution" as opposed to correlation; but there is no doubt about the lack of effectiveness in HER2 targeting for clinical prostate cancer. Interestingly, in the present studies, the authors chose to use pre-clinical models which are actually quite responsive to such targeting. Minimally, additional human lines need to be tested in vivo to allow any possibility of generalizations to the clinic. With regard to the IL-6, the authors report on page 6 that using their chosen PC lines, that: "PC cells with SPRY2 knockdown.... show significant increase in IL-6 expression." They also report that clinical samples of CRPC have down regulation of SPRY2. They also correctly reference, however, that Yu et al 2015 have published that metastatic CRPC cells lack IL-6 production. They also document in Figure 3F that their chosen pre-clinical models respond quite well to IL-6 neutralization, but on page 16 they correctly state that "clinical trials using anti-IL-6 therapy have no benefits." On page 17 the authors propose targeting SR-B1 mediated blockade of tumor cholesterol transport... in hormonally autonomous prostate cancers, but do not robustly test their hypothesis in several lines to validate its generalizability. With regard to the role of HSD3B1 there is also a technical issue. Using the NPS transgenic mouse model, HSD3B1 expression based upon IC staining (i.e. Fig 1F) appears higher in the mock vs. ADT resistant tumors, but Figure 3D reports that in the CWR22 model, ADT increases HSD3B1. Another technical issue is the fact that the level of tumoral testosterone reported for the CWR22 following ADT using ELISA assays is more than a log higher than reported using a much more accurate LS-MS method (i.e. see Titus MA et al PLoS One 2012; 7(1): e30192 doi: 10.1371).

1st Revision - authors' response

3 January 2018

Reviewer #1 (Comments on Novelty/Model System for Author):

Please see remarks to the author regarding the model system.

Reviewer #1 (Remarks for Author):

The manuscript by Patel et al. describes a variety of downstream effects of Sprouty2 (SPRY2) depletion in a mouse model of prostate cancer (PC) engineered for prostate-specific heterozygous loss of Spry2 and Pten as well as in PC orthotopic xenografts and PC cell-based experiments. This study extends previous work from Leung and collaborators (Gao et al EMBO Mol Med 2012), which demonstrated that heterozygous Spry2 and Pten deficiency results in constitutive EGFR/HER2 signaling.

These previous studies further showed that Spry2 deficiency sensitizes PC cells to EGF stimulation in a PTEN-dependent fashion.

The current manuscript describes how SPRY2 depletion leads to:

1. *HER2-mediated PC cell synthesis of androgens and evasion of standard androgen deprivation therapy (ADT),*
 2. *Induction of IL6 and cholesterol metabolism as well as the ramifications of anti-IL6 therapeutic approaches for primary PC vs. metastases and*
 3. *Increased cholesterol scavenger receptor B1 (SR-B1), which increases cholesterol uptake and may serve as a potential therapeutic target in SPRY2 deficient PC tumors.*
While most of the experiments taken individually are carefully conducted and the results sound, the manuscript is very unfocused and lacks some important experimental details. Further, the manuscript lacks a logical flow and is not cohesive making it not only difficult to follow but also hard to evaluate the major conclusions.

Response:

We thank the Reviewer for providing a critical evaluation of our manuscript and constructive criticism. Based on the suggestions, we have substantially revised the manuscript to improve the clarity and focus of our report. We hope the reviewer will now find the revised manuscript more coherent and easier to evaluate. We have made major changes to the entire report as outlined below:

- (i) We have included additional background information in the Introduction section.
- (ii) We have added relevant details on the experimental models, providing important information on the design of the models used (Figure EV2). These experimental details are incorporated into the Methods section (Appendix Supplementary Methods).
- (iii) To enhance the logical flow of the data, we have revised the Results section by changing the order in which data are presented. We also stated specific experimental objectives followed by connecting statements to improve the link from one experimental observation to the next.
- (iv) We have substantially revised the Discussion section, including appropriate consideration in a broad context based on the major findings of this study.

Specific Comments:

1. *The title is extremely misleading in that it is relevant to only approximately half of the manuscript. Similarly, the introduction is incomplete, as it does not mention some of the pathways that are being investigated.*

Response:

We have updated the title to ‘*Sprouty2 loss induced IL6 drives castration-resistant prostate cancer through scavenger receptor B1*’ to emphasise the full impact of our work. The current title is relevant to the entire revised manuscript.

We have edited the introduction to include additional information to introduce all the pathways investigated (Main manuscript text: Page 3).

2. *There is minimal description of the CWR22 cell line used to generate cells depleted of SPRY2. CWR22 is typically propagated serially as mouse xenografts and it is unclear if the authors used the cell line, CWR22Pc, which is derived from CWR22. Also, the pten status should be stated, which is presumably wild type.*

Response:

To improve the clarity and understanding of the CWR22 derived cell lines used for SPRY2 knockdown related experiments, we have now provided the following information in the Methods sections. We have also included additional functional and characterisation data which may help in interpreting the experimental observations presented in this manuscript. We have also added data on PTEN status along with appropriate discussion (Main Manuscript Text: page 18-19).

Briefly, the additional information are listed below:

- (i) CWR22Res cells were used to study the functional effects of SPRY2 knockdown in mediating treatment resistance.
- (ii) CWR22Res cells were obtained from the Case Western Reserve University, Cleveland, Ohio. These cells were maintained in RPMI medium supplemented with 2 mM Glutamine and 10% FBS.
- (iii) The CWR22Res cells are hormone responsive and showed a significant reduction in growth under hormone deprived culture condition (namely 10% Charcoal striped serum/CSS medium) (Fig EV2C).
- (iv) CWR22Rv1 cells were obtained from ATCC (ATCC® CRL-2505™). These cells are maintained continuously in phenol red-free RPMI medium with 2 mM Glutamine and 10% CSS to mimic the condition for castration-resistant prostate cancer (CRPC).

- (v) The Short Tandem Repeat (STR) based DNA profiling analysis was used for cell line authentication (Appendix table 1), confirming that the CWR22Res and CWR22Rv1 cells showed more than 95% match to CWR22Rv1 profile.
- (vi) The growth rate of CWR22Rv1 was unaltered by hormonal levels in the culture condition (Fig EV2C). Since the SPRY2 levels were higher in hormone responsive CWR22Res cells compared to CWR22Rv1 cells (Fig EV1K), the CWR22Res cells were therefore chosen to study the functional effects of SPRY2 (Fig EV1K). CWR22Res and CWR22Rv1 have comparable levels of PTEN and AR (full length and variants). Of note, CWR22Rv1 cells expressed higher levels of HER2, p-p38 and HSD3B1 than the hormone responsive CWR22Res cells (Fig EV3N).
- (vii) CWR22Res cells, when injected in one of the anterior prostate lobes of CD-1 Nude mice, form prostate orthografts with 100% incidence. Implanted CWR22Res cells consistently form palpable orthografts or ultrasound detectable tumours at ~30 days post-surgical implantation. At this stage, the experimental mice were randomized to receive either ADT in the form of castration or Mock intervention with sham surgery (Fig EV2E-F). In ADT treated mice, as expected, the androgen responsive CWR22Res orthografts showed a significant reduction in tumour size after castration compared to Mock-treated orthografts (Fig EV2E).
- (viii) ADT or suppression of SPRY2 expression does not alter PTEN levels in the CWR22Res orthografts (Fig EV2M).

3. The manuscript uses two models with differing pten status. Given that this lab previously identified an important contribution of PTEN to the response of PC in the setting of SPRY2 deficiency, this issue requires examination. Furthermore, the manuscript only includes the Spry2 expression data from patients samples; Pten expression data should also be included.

Response:

We thank the reviewer for raising a valid point of differing PTEN status in the two model systems used in this study and the lack of PTEN expression data in the clinical samples. These points are addressed by new data listed below:

- (i) PTEN expression data from HN and CRPC matched array are now included. Figure EV1B shows that hormone naïve (HN) samples obtained at the time of diagnosis and castration-resistant prostate cancer (CRPC) tumours have comparable PTEN expression. We observe that half of the cases (19 out of 35) had no detectable PTEN immunoreactivity at diagnosis in HN state (Fig EV1B). Thus, PTEN was negligible in 50% of both HN and CRPC cases in our match TMA.
- (ii) In contrast to the PTEN status, SPRY2 expression levels were significantly decreased in CRPC samples when compared to the paired HN cases (Fig 1D). Interestingly, in CRPC patients, SPRY2 levels significantly correlated with PTEN levels (Fig EV1C).
- (iii) CWR22Res cells are PTEN proficient (Fig EV1K), and SPRY2 knockdown does not change PTEN expression in Mock or ADT treated orthografts (Fig EV2M).
- (iv) By transiently knocking down PTEN expression, we show that in hormone deprived conditions (CSS), SPRY2 deficiency offers growth advantage regardless of the status of PTEN (Fig EV2N-O).
- (v) Furthermore, we have characterised the effects of hormone deprivation on SPRY2 status in SPRY2 and PTEN double-deficient LNCaP cells with stable expression of SPRY2 (Fig IH, EV1K, EV2A).

4. IHC analyses throughout the manuscript require some type of scoring or quantification.

Response:

The IHC analyses presented in the manuscript has now been quantified and presented as (Fig EV1G, J, EV3B, EV4K, M, EV5N, Appendix Fig S1E, Appendix Fig S2N and Appendix Fig S3B-D).

5. Although there is significant focus on IL-6, there is little discussion of how the authors' findings fit into the extensive literature on IL-6 in prostate cancer.

Response:

In the revised manuscript we have appropriately discussed how our findings fit into the extensive literature on IL6 in prostate cancer (Main manuscript text: Pages 20-23).

6. There is a differential effect of ADT in the NPS GEMM mouse PC model at 1 month vs 6 months ADT but there is no discussion of the timing of ADT in the orthotopic model nor how the two models compare.

Response:

We have now included the following information in the revised manuscript: To investigate the role of SPRY2 in mediating treatment-resistance, we carried out a pilot experiment where approximately 14 million cells (Nsi control or SPRY2 deficient clones –CL3 and Pool) were injected in one of the anterior prostate lobes of CD-1 nude mice. For this experiment, we used ten mice for each of the cell clone being injected. As observed with parental CWR22Res cells, irrespective of SPRY2 status, orthografts were palpable around 30 days with 100% incidence. These mice were then randomised to receive either Mock or ADT treatment (n=5 per treatment). All Mock-treated animals achieved maximum permitted tumour burden around 73 days post-implantation irrespective of SPRY2 status (Fig EV2H). Following ADT (castration), mice with Nsi orthografts survived longer than Mock-treated mice. In contrast, ADT in mice harbouring SPRY2 deficient orthografts resulted in signs of weight loss around 60 days (Fig EV2H-I). The ADT treated mice with SPRY2 deficient orthografts reached clinical endpoints between 60-70 days (Fig EV2H). Hence, for subsequent experiments with the orthograft model, we used a refined 60 days timed experimental protocol to carry out detailed investigations. Since the tumour incidence was 100%, we used n=5 per group in all our timed experiments. Hence, all the experimental data based on orthografts (+/- treatments) presented thereafter were generated using the 60 days timed protocol.

7. It is not clear whether intra-tumoral testosterone levels were quantified by ELISA or LC-MS.

Response:

We have used both ELISA and LC-MS. The ELISA kit (Cayman Chemicals) was used to measure the tumoral testosterone levels in Figure 2E. The relative levels of androstadiene and testosterone in orthografts presented in Figure 6D-E were obtained using LC-MS. We have clarified this in the legends and the respective sections in the Method section.

8. The previous paper (Gao et al.) examined the effects of SPRY2 loss in androgen receptor negative PC cells but the current work focuses on an androgen receptor (and possibly androgen receptor splice variant) expressing model yet there is essentially no analysis of the contribution of androgen receptors.

Response:

We agree with the comment raised by the Reviewer. We have now examined AR levels using a well-documented anti-AR antibody (ARN20 sc-816 from Santa Cruz Biotechnology) that detects full length and variants (PMID: 20823238). We have included the following new data in the revised manuscript:

- (i) The levels of AR (full length and variant) are comparable between the SPRY2 knockdown and Nsi control CWR22Res orthografts from mock treated animals. Compared to the mock treated Nsi control CWR22Res orthografts, the levels of AR (full length and variant) are dramatically reduced in ADT treated Nsi orthografts. SPRY2-deficient prostate orthografts however maintained the levels of AR (full length and variants) following ADT (Fig 2A).
- (ii) SPRY2 over-expression in LNCaP cells does not change AR levels (Fig 2I).
- (iii) All treatments that sensitised SPRY2-deficient orthografts to ADT, namely HER2 knockdown, Tocilizumab, Simvastatin and ITX5061, also decreased the levels of AR (Fig 2N, Fig EV4I, Appendix Figure S2O).

9. The pooled CWR22 vs CL3 cells exhibit different extent of SPRY2 depletion. How might this difference contribute to the regulation of CYP17A1 that is only found in the pooled SPRY2-knock down cells? Similarly, overexpression of SPRY2 in LNCaP cells results in increased CYP17A1 and HSD3B1. These results suggest that the focus solely on HSD3B1 may not be justified.

Response:

We appreciate the critical observations made by the Reviewer, and agree with the point raised. We have now included the following additional data and appropriate explanation in the Results section of the revised manuscript:

The CYP17A1 levels in the CWR22Res orthografts remain unaltered irrespective of treatment or SPRY2 levels (Fig EV3E). On the other hand, SPRY2 deficiency was consistently associated with elevated HSD3B1 levels in all of the following CRPC models used in this study:

- (i) SPRY2 deficient CWR22Res orthografts showed increased expression and protein levels of HSD3B1 following ADT (Fig 2A, 2H, 2N, EV3C, EV3D).

- (ii) In LNCaP cells, overexpression of SPRY2 dramatically decreased the protein levels of HSD3B1 (Fig 2I).
 - (iii) The CRPC tumours from NPS mice also showed high HSD3B1 levels compared to mock treated mice (EV3F).
 - (iv) The CRPC orthografts generated by growing VCaP (naturally SPRY2 deficient, PTEN and AR-full length and variant-proficient) cells in castrated mice showed higher HSD3B1 protein levels compared to the orthografts from uncastrated mice (Fig EV3H).
- Thus based on these observations, we hypothesised that HSD3B1 may offer a survival advantage in SPRY2 deficient tumours under ADT stress.

10. *Experiment in figure 3F lacks intact mice as controls.*

Response:

This information on the effects of anti-IL6 treatment on Mock treated NPS mice is presented as Fig 3I.

11. *The manuscript would be strengthened by stating the objectives of the study more clearly, improving the logical flow of data presentation and removing extraneous data.*

Response:

We appreciate the advice and have now extensively revised our manuscript to address this point. The changes are summarised below:

- (i) The objectives of the study are highlighted in the revised manuscript.
- (ii) Relevant background information is added to the Introduction section, along with more information on the experimental objectives in the Results section.
- (iii) The figures and results sections have been extensively edited to improve the flow of the report. Utilising the expanded view figures, we have added relevant supporting information to better explain the work presented in the main figures.
- (iv) Extraneous data that add no or little meaning to the current work presented have been removed or move to Appendix figure S3.

12. *It appears that ITX is no better than statins with respect to androstenedione or testosterone levels as compared under ADT conditions. Thus, it is not clear why the authors conclude that SR-B1 represents an exciting new target for PC. The authors should address this issue experimentally.*

Response:

The Reviewer has correctly pointed out that ITX5061 and Simvastatin treatments *in vivo* decreased the levels of androstenedione and testosterone in the orthografts to a similar extent. Our work shows that these two drugs may act at two different nodes during disease progression to sensitise tumours to ADT. We identified these two nodes by studying the development of CRPC at a tumour intrinsic and extrinsic level. We have now clarified this point in the Discussion section (Main manuscript text: Page 24-25).

At a tumoral level, compared to mock treated orthografts, ADT treated orthografts (irrespective of SPRY2 levels) have reduced expression of HMGCR, a rate-limiting enzyme for cholesterol biosynthesis and a simvastatin target (Fig 4A). Instead, the increase in tumoral cholesterol may be due to increased expression of SRB1, HDL cholesterol transporter and ITX5061 target (Fig 5E). Hence, ITX5061 may act at the tumour level by blocking the cholesterol uptake into CRPC cells. This limited cholesterol uptake may be an important contributor to the reduced levels of androstenedione and testosterone within ITX5061 treated orthografts under ADT conditions. At a systemic level, tumour-induced IL6 modulates adipose lipolysis and hepatic cholesterol metabolism. The elevated serum cholesterol may be a consequence of systemic effects of IL6 which leads to elevated expression of hepatic HMGCR (Fig 5A). Hence, statin was effective to normalise the systemic cholesterol homeostasis. The effects of statin treatment seen on the orthografts are likely due to decreased levels of hormone synthesis (as seen by the levels of androstenedione and testosterone) due to limited systemic cholesterol bioavailability rather than direct cytotoxic effects on orthografts.

Hence, based on our observation, statins may act on the hepatic cholesterol biosynthesis, while ITX5061 may (more specifically) target tumoral cholesterol uptake.

In the current revised manuscript, we have experimentally investigated the direct growth suppressive effects of simvastatin and ITX5061 (Appendix Figure S3F). Overall, in both LNCaP and

CWR22Res derived cells, SPRY2 deficient cells are much more sensitive to ITX5061 treatment. However, due to the differing IC50s of the drug in different cells, it may not be possible to make such direct comparison of efficacy.

Hence, we have edited our conclusion to ‘Thus, prostate patients with SPRY2 loss or HER2 activation may progress to form androgen autonomous AR activated CRPC. This type of treatment-resistant cancer may respond to treatments targeting systemic cholesterol bioavailability (statins) and tumoral cholesterol transport (SRB1 antagonist, ITX5061)’.

13. Analysis of the effects of abiraterone should be conducted in at least one of the two SPRY2-deficient model systems.

Response:

We thank the Reviewer for suggesting this. We have tested the effects of abiraterone in CWR22Res cells with differing levels of SPRY2. We are now able to make the following changes to the manuscript.

SPRY2 deficient CWR22Res cells were more resistant to abiraterone in hormone deprived conditions as compared to the Nsi control cells (Fig EV3J). Importantly, while HSD3B1 knockout and abiraterone treatment independently inhibited the growth of SPRY2-deficient cells to a similar extent under hormone deprived conditions, the combination treatment (HSD3B1 knockout and abiraterone) was more effective in decreasing the growth of SPRY2 deficient cells (Fig 2J). These results may suggest that HSD3B1 and CYP17A1 dual inhibition may cooperate to mitigate growth under ADT conditions. Consistent with our observations, HSD3B1 protein stabilising mutations have been shown to arise under abiraterone therapeutic stress (PMID: 23993097). Furthermore, resistance to abiraterone may be mediated by induction of CYP17A1, AR or AR variants (PMID: 21807635). AR activation mediated by intra-tumoral steroid biosynthesis may also mediate abiraterone resistance (PMCID: PMC3209585). Thus, a subset of abiraterone resistant CRPC may continue to retain steroid biosynthesis. Our current work on (i) limiting the tumoral cholesterol uptake (using SRB1 antagonist e.g. ITX5061) and (ii) restoring the systemic cholesterol homeostasis (using statins) may suggest that targeting a parallel node of cholesterol bioavailability may aid in sensitising the tumours to ADT.

Reviewer #2 (Comments on Novelty/Model System for Author):

The authors have used multiple mouse models, including CRISPR knockout and shRNA knockdown cell line in vivo models making it a very comprehensive and quality technical analysis of the topic. While each individual component is not necessarily novel (see comments on related publications that require citing and discussion), the data convincingly link these signalling pathways together, which makes it a novel study with novel findings. Considering the medical implications of repurposing currently approved drugs such as statins or the SR-B1 inhibitor, this makes the medical impact high.

Response:

We thank the Reviewer for the general appreciation of our work.

Reviewer #2 (Remarks for Author):

While the authors have undertaken a comprehensive experimental analysis of the links between RTK signalling, SPRY2, IL6 and cholesterol uptake through SRB1 in prostate cancer, they have neglected to appropriately reference and discuss a number of key publications in SPRY2 and PCa papers (including one of their own PMID: 26075267, PMID: 23150596 and PMID: 23584380), IL-6, statins and PCa (PMID: 24296978) and SR-B1 in PCa (PMID: 19866465, PMID: 18782595 and PMID: 22025344). While these do not detract from the experimental data presented, these should be included and discussed as important precursors for their work - which nicely links these previous findings together.

Response:

We thank the Reviewer for pointing out these important publications. We have now appropriately cited these reports (Main manuscript text: Pages 4, 22, 24). We agree with the Reviewer that the additional citations help to provide a better perspective on our work.

The authors have done a good job of profiling the effects of SPRY2 deficiency on response to ADT, both in a GEMM het SPRY2/PTEN model, as well as in a cell line with SPRY2 KD. These data convincingly show that SPRY2 loss leads to resistance to ADT. Addition of PTEN status to Figure S2C would be helpful, since the GEMM mouse model combines PTEN and SPRY2 heterozygosity. This is particularly important as the authors have published that CWR22 cells have intact PTEN - and indeed that SPRY2KD does not substantially alter PTEN expression in this same CL3 line. This should be explicitly stated and discussed in the text, as it is a clear difference between the GEMM and the orthograft model.

Response:

We agree with the Reviewer on the differences in the PTEN status of our models used and the importance of indicating these differences. We have now included the data on PTEN status as Fig EV1K and EV2M. ADT or SPRY2 deficiency does not change PTEN levels in CWR22Res orthografts. We have also adequately discussed the differences in the models used here (Main manuscript text: Page 19).

Since the authors have a good antibody for HSD3B1, it would have been more convincing to perform western blots to confirm qPCR expression in FigS3B, S3D and Fig2B-C and 5I.

Response:

We have now performed Western blot analysis for HSD3B1 as requested and have presented these as follows:

- (i) Fig 2A shows the protein levels of HSD3B1 in Mock and ADT treated orthografts from Nsi control and SPRY2 knockdown (clones CL3 and Pool) cells. Compared to Nsi control, SPRY2 deficient orthografts showed increased HSD3B1 levels in ADT treated mice.
- (ii) Fig 2I shows the HSD3B1 protein levels in control and SPRY2 expressing LNCaP cells. Compared to the control cells, HSD3B1 levels were lower in SPRY2 expressing LNCaP cells.
- (iii) Fig 2N shows the levels of HSD3B1 in Mock and ADT treated Nsi control and SPRY2 deficient orthografts with the HER2 knockdown. HER2 knockdown decreased the levels of HSD3B1 in ADT treated SPRY2 deficient orthografts.
- (iv) Fig EV3D shows the HSD3B1 levels in Nsi control and SPRY2 knockdown clones (CL3 and Pool) grown in 2-dimensional culture conditions. SPRY2 deficient clones show higher levels of HSD3B1 compared to Nsi control CWR22Res cells.
- (v) Fig EV3F shows the HSD3B1 levels in Mock and ADT treated NPS tumours. ADT treated NPS tumours have higher levels of HSD3B1 compared to tumours from Mock-treated mice.
- (vi) Fig EV3K shows the HSD3B1 levels in Nsi control and SPRY2 knockdown (Pool) cells with the HSD3B1 knockout.

The authors have nicely shown that HER2 KD can block IL6 expression. Since "SPRY2 deficiency may confer androgen self-sufficiency by inducing HER2 mediated IL6 cytokine axis", correlating changes in HER2 (Fig S3F) and IL6 (Fig 2L) would assist in supporting these data.

Response:

We thank the Reviewer for this suggestion, and have correlated the immunoreactivity of IL6 and HER2 expression in our match TMA and incidence array. We have included the following text in the revision (Main manuscript text: Page 10):

‘In clinical prostate cancer samples, HER2 levels significantly correlated with IL6 levels ($r = 0.2446$; $p = 0.0288$; $n = 80$).’

An interesting finding is that anti-IL6 reduces tumour burden, but increases metastasis. Do the CRPC patients with increased serum IL-6 have reduced metastasis (from Figure 4B)?

Response:

We thank the Reviewer for pointing out one of the interesting findings of our work. From data available from our patient cohort, we have analysed the metastatic incidence rate in relationship to serum IL6 and can make the following observations:

- (i) We compared the serum IL6 levels between patients with and without clinical/radiological evidence of metastasis (M0 and M1 respectively). We found no association between serum IL6 levels and metastasis status.

(ii) We further focused our analysis in the patients with CRPC with information on metastases (n=58). Stratifying the CRPC sub-cohort by the median serum IL6 level, the metastasis status was not found to associate with IL6 levels (Chi-square test, $p=0.09$; M0/M1: 21/8 for low IL6 and 26/3 for high IL6, respectively).

Overall, in the cohort we have analysed, we did not find an association between serum IL6 and clinical evidence of metastatic disease.

Why is Figure 4E 42 patients and Figure 4F only 18 patients?

Response:

For assaying levels of IL6, we requested the serum samples from the ProMPT study (ethics committee approval: UK MREC number 01/4/61) (PMID: 22240788). The samples obtained (172 patients with HNPC and 129 patients with CRPC) were enough to perform IL6 ELISA with technical replicates per sample. The patient records were used to acquire the data on PSA levels. Our criteria for IL6 and PSA correlation analyses were to obtain the PSA level data on the same serum samples (matched for sample collection date) from individual patients. Only 42 patients matched these criteria, and in the revised manuscript this is presented as Fig 4G.

For the free fatty acid (FFA) measurements performed in our laboratory, we wanted to apply same criteria. For this, we requested a subset of 20 serum samples from the ProMPT study, randomly selecting 10 HNPC and 10 CRPC cases respectively that we already obtained data on their IL6 levels. We could successfully assay 18 of the newly acquired samples (allowing for adequate material for technical duplicates). Hence, the FFA and IL6 correlation data presented in the new Fig 4N (previously Fig 4F) is for a subset of 18 samples.

The effects of IL6 on lipolysis and serum lipid species is nicely shown.

Response:

We thank the Reviewer for appreciating our efforts.

Figures S8C and D need some statistical analysis. More explanation of the IC50 analysis undertaken in S8F should be provided. It is hard to interpret the data currently supplied as the range of doses used is not wide enough and there are no details on what is the control in each case (compared to DMSO with parent or KO cell line, FM or CSS?). It is hard to determine how the IC50 data were generated from these curves - and indeed there are no IC50 data for CWR cells.

To address the points raised by the Reviewer, we have done the following:-

(i) We have provided statistics for original figure S8C which is now Fig EV5J. We performed two way ANOVA with Tukey's test. [* indicates $p<0.05$ compared to DMSO treated Nsi control (far left blue bar), and # indicates $p<0.05$ compared to ITX5061 treated Nsi control (far left red bar)]. We have included the appropriate details in the figure legends.

(ii) The data presented in Figure S8D (now is EV5E) is not significantly different ($p=0.06$). Such variability is possibly due to the high variability we have experienced while working with LNCaP AI cells. We have assessed HDL and LDL uptake in VCaP cells cultured in hormone proficient (FBS) or deficient (CSS) conditions (Fig EV5F). In VCaP cells, HDL uptake was significantly increased in hormone deficient condition.

(iii) The cytotoxic effects of ITX5061 were assayed using WST-1 assay in CWR22Res derived control and SRB1 KO cells cultured for 48 hr in RPMI with 10% charcoal stripped serum (CSS) mimicking hormone deprived (ADT) conditions. Dose response of ITX5061 was analysed relative to respective DMSO. IC50 was calculated using log (inhibitor) vs. response -- Variable slope (four parameters) with Bottom constraint=0.0 with GaphPad Prism software. IC50 of ITX5061 in CWR22Res control= $23.87 \pm 0.677 \mu\text{M}$ and CWR22Res SRB1KO = not calculated due to not convergent data or no dose response observed. These data suggested that ITX5061 mediated its cytotoxic effects in presence of SRB1. We have included the detailed information in the expanded view figure legends (EV5C).

The Main text, figures and Supplementary data have many mistakes, and should be thoroughly edited for English, spelling, and typographical errors. For example, spelling of Sprouty2 is incorrect on line 57 page 6, "self-sufficient from of CRPC" on line 304 page 13, and there are lots of other inconsistencies throughout the main figures (e.g SRPY2 in Figure 6C and D; use of "u"

instead of micro in multiple figures etc) and in the supplementary figures (RMPI instead of RPMI, Figure S8 legend says Figure S7, ug and ul instead of micro symbol, consistency of labelling etc).

Response:

We regret the number of unnecessary errors (in the text and figures) in the initial submission, and would like to thank the Reviewer for pointing them out. In the revised manuscript, we have paid careful attention to annotations and spellings.

There is no Figure 7F despite being referred to in the text "Furthermore, the SR-B1 expression at diagnosis also correlated significantly with overall survival in ADT treated patients (Fig. 7F)."

Response:

We apologise for these errors. We have now checked that the correct figures are cited in the text.

Figure S9C may be more helpful in the main text, as this assists in bringing together the findings in the paper.

Response:

We thank the Reviewer for this suggestion. In agreement, we have included this piece of data as a main figure panel in Fig 7D.

Reviewer #3 (Comments on Novelty/Model System for Author):

The majority of the human xenograft data was generated using only the CWR22 line. Additional lines, particularly the LNCaP should be similarly tested since it has low SPRY2 expression coupled with significant expression of HER2.

Response:

We appreciate the Reviewer's suggestion of extending our study to other models such as LNCaP cells. We have previously (PMID: 23434594) shown that compared to control LNCaP cells, LNCaP cells with SPRY2 expression have less tumorigenic potential and form negligible prostate orthografts. Hence, in the revised manuscript we have included the following *in vitro* data on the effects of hormone depletion and SPRY2 levels in LNCaP cells and appropriately discussed these observations:

- (i) 'SPRY2 overexpression significantly decreased the growth of LNCaP cells in hormone proficient conditions (FBS) (Fig 1H, EV2A). Importantly, the SPRY2 overexpression induced inhibition of cell growth was significantly more profound under hormone deprived conditions (CSS) (Fig 1H).'
- (ii) SPRY2 overexpression significantly decreased the IL6 expression in LNCaP cells (Fig EV4B).
- (iii) ITX5061 reduced the growth of LNCaP cells in hormone deficient conditions (Fig EV5K).

Reviewer #3 (Remarks for Author):

There are several issues with this manuscript, the most significant is whether the results are generalizable to the clinic.

The authors state on page 15 that: "Despite the overwhelming clinical occurrence and functional contributions of HER2/RTK signaling in driving treatment resistance in prostate cancer, inhibition targeting RTK's ... have offered limited success." One can argue about the statement about "overwhelming contribution" as opposed to correlation; but there is no doubt about the lack of effectiveness in HER2 targeting for clinical prostate cancer.

Interestingly, in the present studies, the authors chose to use pre-clinical models which are actually quite response to such targeting. Minimally, additional human lines need to be tested in vivo to allow any possibility of generalizations to the clinic.

Response:

We appreciate the Reviewer's pertinent consideration concerning the generalizability of our results to the clinic. In the revised manuscript, we have appropriately discussed how our report relates to clinically aggressive prostate cancer. We appreciate the consideration by the Reviewer in performing experiments using additional preclinical models. With the constraint of the time available for preparing for a revised manuscript, it is not possible to formally carry out additional *in*

in vivo studies. Hence, we have discussed the importance of further investigating the role of SRB1 and efficacy of ITX5061 in other relevant *in vivo* xenografts model systems using C4-2 CRPC variant of LNCaP and VCaP cells.

Also, we have now included data on effects of ITX5061 on the growth of CWR22RV1 cells (the CRPC variant of CWR22) (Fig EV5L).

With regards to the HER2 related discussion, we thank the Reviewer for the comment and have rephrased this section as below:

‘HER2 and the downstream signalling cascades are potentially clinically actionable targets and may play a functional role in driving treatment-resistance (Robinson et al, 2015; Roychowdhury & Chinnaiyan, 2016; Shiota et al, 2015). In early phase I and II clinical trials, while HER/EGFR inhibitors do not alter the disease progression in untreated hormone naïve patients, a subset of CRPC patients treated with Lapatinib, a HER2/EGFR inhibitor, did show a decrease in disease progression (Sridhar et al, 2010; Whang et al, 2013). Likewise, our work underscores the importance of HER2 in promoting tumour growth only under ADT condition.’

With regard to the IL-6, the authors report on page 6 that using their chosen PC lines, that : " PC cells with SPRY2 knockdown.... show significant increase in IL-6 expression." They also report that clinical samples of CRPC have down regulation of SPRY2. They also correctly reference, however , that Yu et al 2015 have published that metastatic CRPC cell lack IL-6 production. They also document in Figure 3F that their chosen pre-clinical models respond quite well to IL-6 neutralization, but on page 16 they correctly state that "clinical trials using anti-IL-6 therapy have no benefits."

Response:

We appreciate the contradictions indicated by the Reviewer. To improve clarity and understanding of IL6 in CRPC, we have included the following revisions:

- (i) Consistent with increased IL6 expression in SPRY2 deficient CWR22Res cells, SPRY2 expressing LNCaP cells showed lower IL6 expression compared to control cells (Fig EV4B).
- (ii) The human androgen-independent DU145 and CWR22RV1 cells showed higher IL6 expression compared to the androgen-dependent LNCaP and CWR22Res cells (Fig EV4F).
- (iii) In clinical CRPC patients, SPRY2 expression inversely correlated with IL6 immunoreactivity in the prostate tumours ($r = -0.427$; $p=0.0261$; $n=27$; Fig EV4E).

Collectively, based on data presented in figures 3 and EV4, SPRY2 deficiency is associated with upregulated IL6 expression in CRPC. Based on our observations, it is likely that in clinical prostate cancer, patients with SPRY2 deficiency may show enhanced tumoral IL6 levels. SPRY2 loss induced IL6 may mediate treatment resistance.

Our work shows that CRPC cases have higher IL6 levels compared to paired HN tumour biopsies (Fig 3D). A number of studies suggest an association of the autocrine IL6 function in experimental models of prostate cancer and CRPC (PMID: 25374925). Our work may indicate that molecular events such as SPRY2 loss or HER2 activation may govern the autocrine role of IL6 in mediating CRPC. However, a recent study by Yu et al (PMID: 26048576) reported little evidence of IL6 expression in the prostate tumours and metastases collected either during radical prostatectomy or autopsy. Hence, additional studies that are adequately powered are required to further characterise the status of IL6 expression in CRPC samples.

Although tumoral IL6 production may be debatable, IL6 receptor is present uniformly in majority of prostate cancers. In addition to autocrine role of IL6, we also provide evidence of elevated IL6 expression from epididymal adipose tissue and higher serum IL6 levels in mice bearing CRPC orthografts (Appendix figure S2C). Systemic IL6 may also contribute to tumoral expression of HSD3B1 and SRB1 to mediate treatment resistance. It is worth noting that systemic IL6 is also clinically important, with high serum IL6 levels confer a poor survival outcome in CRPC patients (Fig 4H).

In the revised manuscript we have discussed the pointed raised by the Reviewer as follows:

‘IL6 being a pleiotropic cytokine may arise from multiple sources in cancer patients. While reports indicate little IL6 production from primary or metastatic clinical prostate cancer samples, the IL6 receptor is present uniformly in a majority of prostate cancers (Culig, 2014; Siegall et al, 1990; Yu et al, 2015). Importance of molecular events such as SPRY2 loss or HER2 activation in tumoral IL6 production in clinical prostate cancers needs further investigation. The tumour microenvironment including infiltrating immune cells and stromal tissue including prostate cancer-associated fibroblasts may also substantially contribute to tumoral IL6 (Doldi et al, 2015).

The value of targeting the IL6 pathway in prostate cancer including CRPC remains unclear (Culig, 2014). While IL6 targeting therapies have not demonstrated clear effects on clinical outcomes such as progression free survival, there appeared to be a subgroup of patients that demonstrated good PSA response as well as clinical and radiologic evidence of stable disease (Dorff et al, 2010; Fizazi et al, 2012).’

On page 17 the authors propose targeting SR-B1 mediated blockade of tumor cholesterol transport... in hormonally autonomous prostate cancers, but do not robustly test their hypothesis in several lines to validate its generalizability.

Response:

We thank the Reviewer for raising this point. By knocking out *SRB1* in CWR22Res cells, we show that ITX5061 decreases growth rate of SPRY2 deficient CWR22Res cells by decreasing HDL uptake (Fig 6A-B, EV5A-D, J). We have included the following in the revised manuscript to validate our hypothesis:

- (i) As observed in CWR22Res cells, ITX5061 (*SRB1* antagonist) treatment also significantly decreased HDL uptake in LNCaP, VCaP and CWR22RV1 cells (Fig EV5 G-I).
- (ii) As observed in CWR22Res cells, ITX5061 treatment significantly reduced growth rates of LNCaP and CWR22RV1 cells in (Fig EV5K-L)

We have also added the following to the Discussion section:

‘Our data support the repurposing of ITX5061 in treating CRPC. ITX5061 treatment re-sensitised prostate tumours to ADT by inhibiting *SRB1* mediated cholesterol uptake, thereby decreasing tumour androgen biosynthesis. Further investigations are therefore warranted to robustly examine the effectiveness and safety of *SRB1* antagonist such as ITX5061 to tackle treatment resistant prostate cancer. Additional pre-clinical models will be required to identify suitable patient groups and to define the optimal treatment schedule in using ITX5061.’

With regard to the role of HSD3B1 there is also a technical issue. Using the NPS transgenic mouse model, HSD3B1 expression based upon IC staining (i.e. Fig 1F) appears higher in the mock vs. ADT resistant tumors, but Figure 3D reports that in the CWR22 model, ADT increases HSD3B1.

Response:

We have now further investigated the point of HSD3B1 IHC staining in the NPS tumours. While HSD3B1 IHC staining was patchy in mock treated NPS tumours, the staining was consistent, and uniform in ADT treated tumours. Hence to address the technical point of HSD3B1 abundance in mock vs ADT NPS tumours, we have examined the tumour levels of HSD3B1 by Western blotting. Based on this, we can conclude that HSD3B1 levels are higher in CRPC NPS tumours compared to mock-treated tumours (Fig EV3F). We have also complemented RT-PCR data from orthografts with Western blots. We found that HSD3B1 levels are much higher in CRPC orthografts (Fig 2A, 2N, EV4I). Similarly, CRPC VCaP orthografts also show elevated protein levels of HSD3B1 (Fig EV3H).

Another technical issue is the fact that the level of tumoral testosterone reported for the CWR22 following ADT using ELISA assays is more than a log higher than reported using a much more accurate LS-MS method (i.e. see Titus MA et al PLoS One 2012; 7(1): e30192 doi: 10.1371).

Response:

We appreciate the technical point raised by the Reviewer. The differences between our observations and reported observations may be attributed to the following differences between the two studies:

- (i) The CWR22 derived variant used in this study.
- (ii) The prostate orthograft model system instead of subcutaneous xenograft.
- (iii) Timed experimental design instead of subcutaneous tumour cell injection in castrated mice followed by ectopic testosterone pellets.
- (iv) The sensitivity of the ELISA kit (Cayman Chemicals): Although the kit is designed to detect testosterone, it can also identify other androgen species.

We agree with the Reviewer that the LC-MS method is a superior method to detect steroid. For this, we have optimised LC-MS based detection of tumoral steroids and present new data as Fig 6D-E. The sample extraction process including the solid phase extraction may result in loss of certain steroids. Hence, we have presented the data as relative values to the fortified internal standards per mg of orthograft tumour use.

2nd Editorial Decision

26 January 2018

Thank you for the submission of your revised manuscript to EMBO Molecular Medicine. We have now received the enclosed reports from the referees that were asked to re-assess it. As you will see the reviewers are now globally supportive and I am pleased to inform you that we will be able to accept your manuscript pending a few final amendments:

1) Please address both referees' comments in writing. At this stage, we'd like you to discuss referee's 1 points and if you do have data at hand, we'd be happy for you to include it, however we will not ask you to provide any additional experiments at this stage.

***** Reviewer's comments *****

Referee #1 (Remarks for Author):

The authors have substantially improved the manuscript and carefully addressed most of the reviewer's comments. Notably they have re-organized and added relevant background information to the Introduction and Results sections that allows a better understanding of the rationale behind the experimental design and the data analysis. However, there are two experimental aspects that should be considered:

(1) Given that in ADT conditions AR (full length and variants) levels were not affected in SPRY2 depleted models (neither in CWRR22es cell line or SPRY2-deficient orthografts); and that the authors proposed that SPRY2 deficiency mediates ADT resistance and CRPC progression with active AR, it should be analyzed not only AR (full length and variants) protein levels but also AR transcriptional activity under the conditions that sensitized SPRY2-deficient models to ADT.
(2) The authors hypothesize that SPRY2 deficiency promotes CRPC with active AR by modulation of the androgen biosynthetic pathway, mainly through the enzyme HSD3B1. Although, authors showed how SPRY2 depletion increased HSD3B1 expression and protein levels, analysis of the enzymatic activity of the main members of the pathway (CYP17A1, HSD17B1 and HSD3B1) would be necessary to accurately understand the role of this pathway on SPRY2-deficient models of CRPC progression.

Referee #2 (Comments on Novelty/Model System for Author):

Appropriate models are used.

Referee #2 (Remarks for Author):

The authors have adequately answered my questions, however the response to a few of the questions should be incorporated into the manuscript, rather than just being answered in the rebuttal (see below).

"An interesting finding is that anti-IL6 reduces tumour burden, but increases metastasis. Do the CRPC patients with increased serum IL-6 have reduced metastasis (from Figure 4B)?"
Despite the analysis not supporting their data, there are many reasons that could be the case which could be discussed.

"Why is Figure 4E 42 patients and Figure 4F only 18 patients?"
Again, this response should be incorporated into the manuscript to explain to the readers why subsets of samples were used. For example this could be in either the methods or in the Figure legend for Figure 4G and Figure 4N.

2nd Revision - authors' response

9 February 2018

Referee #1 (Remarks for Author):

The authors have substantially improved the manuscript and carefully addressed most of the reviewer's comments. Notably they have re-organized and added relevant background information to the Introduction and Results sections that allows a better understanding of the rationale behind the experimental design and the data analysis. However, there are two experimental aspects that should be considered:

(1) Given that in ADT conditions AR (full length and variants) levels were not affected in SPRY2 depleted models (neither in CWRR22es cell line or SPRY2-deficient orthografts); and that the authors proposed that SPRY2 deficiency mediates ADT resistance and CRPC progression with active AR, it should be analyzed not only AR (full length and variants) protein levels but also AR transcriptional activity under the conditions that sensitized SPRY2-deficient models to ADT.

Response:

We thank the Reviewer for appreciating our efforts. In this manuscript, Tocilizumab, Simvastatin and ITX5061 treatments sensitise the primary prostate orthografts to ADT. We have shown in figure 3 that Tocilizumab treatment that sensitised prostate orthografts to ADT (Fig 3E) have significantly lower AR levels (Figure EV4I) and transcriptional activity based on PSA expression (Figure 3G). Since the AR levels are significantly lower in prostate orthografts from mice treated with a combination of ADT with either Simvastatin (Appendix figures S2M, S2N and S2O) or ITX5061 (Figures EV5M, EV5N and EV5O), the potential transcriptional activity of AR is likely to be lower as well. However, we do agree with the reviewer that validating the AR transcriptional activity may strengthen this work further. This point is now appropriately discussed on page 21: 'Future characterisation of the AR transcriptional activity under the conditions that sensitised SPRY2-deficient models to ADT may provide further understanding of AR reactivation in CRPC.'

(2) The authors hypothesize that SPRY2 deficiency promotes CRPC with active AR by modulation of the androgen biosynthetic pathway, mainly through the enzyme HSD3B1. Although, authors showed how SPRY2 depletion increased HSD3B1 expression and protein levels, analysis of the enzymatic activity of the main members of the pathway (CYP17A1, HSD17B1 and HSD3B1) would be necessary to accurately understand the role of this pathway on SPRY2-deficient models of CRPC progression.

Response:

We agree with the reviewer on the relevance of measuring the enzymatic activities of CYP17A1, HSD17B1 and HSD3B1 in SPRY2 deficient models of CRPC progression. It is indeed interesting to consider if SPRY2 loss mediated receptor tyrosine kinase signaling may affect the overall activity of the androgen biosynthetic enzymes in addition to their relative abundance, perhaps by post-translational modification such as phosphorylations. Interestingly, in the SPRY2 deficient prostate orthografts independent of ADT treatment, we observed an accumulation of Androstenedione, a product of CYP17A1 and HSD3B1 activities (Figure 6D). We have now included the following statement in discussion (on page 22) to cover the point raised by the reviewer: 'In addition to the expression of CYP17A1, HSD3B1 and HSD17B1, analyses of their enzymatic activities in SPRY2 deficient cells may further aid in ascertaining the functional contributions of these androgen biosynthetic enzymes in CRPC progression.'

Referee #2 (Comments on Novelty/Model System for Author):

Appropriate models are used.

Referee #2 (Remarks for Author):

The authors have adequately answered my questions, however the response to a few of the questions should be incorporated into the manuscript, rather than just being answered in the rebuttal (see below).

*"An interesting finding is that anti-IL6 reduces tumour burden, but increases metastasis. Do the CRPC patients with increased serum IL-6 have reduced metastasis (from Figure 4B)?"
Despite the analysis not supporting their data, there are many reasons that could be the case which could be discussed.*

Response:

We have now included the above consideration in the discussion section on page 24 as: 'Despite evidence of increased metastases in animals treated with IL6 neutralising therapies, we did not find any correlation between serum IL6 in patients with prostate cancer and overall metastatic burden. This lack of association can be attributed to a number of factors including the pleiotropic nature of IL6, multiple IL6 sources and systemic inflammation that is associated with widespread metastases.'

"Why is Figure 4E 42 patients and Figure 4F only 18 patients?"

Again, this response should be incorporated into the manuscript to explain to the readers why subsets of samples were used. For example this could be in either the methods or in the Figure legend for Figure 4G and Figure 4N.

Response:

We agree with the reviewer and have incorporated this in the methods sections (Page 28-29). 'Serum levels of PSA, FFA and IL6 levels were measured in the same serum samples from individual patients. The patients' records were used to acquire the data on PSA levels. The study criteria for IL6 and PSA correlation analyses were to obtain the PSA level data on the same serum samples (matched for sample collection date) from individual patients. 42 patients matched these criteria. Hence, IL6 and PSA correlation analysis was performed in 42 samples. For FFA and IL6 correlation analyses, a subset of 20 serum samples was requested from the ProMPT study, randomly selecting 10 HNPC and 10 CRPC cases for which IL6 levels were previously assayed. Out of these 20, 18 samples were analysed for FFA levels. Hence, the FFA and IL6 correlation data presented is for 18 samples'.

Corresponding Author Name: Rachana Patel and Hing Y Leung

Manuscript Number: EMM-2017-08347-V2